# Unraveling the role of the mitochondrial one-carbon pathway in undifferentiated thyroid cancer by multi-omics analyses

Seong Eun Lee [1,2,22], Seongyeol Park [3,21,22], Shinae Yi [1,2], Na Rae Choi [1,2], Mi Ae Lim[2,4], Jae Won Chang[4], Ho-Ryun Won[4], Je Ryong Kim[5], Hye Mi Ko[5], Eun-Jae Chung[6], Young Joo Park [7], Sun Wook Cho[7], Hyeong Won Yu [8], June Young Choi[8], Min-Kyung Yeo[9], Boram Yi[3], Kijong Yi[3], Joonoh Lim[3], Jun-Young Koh [3], Min Jeong Lee[10], Jun Young Heo [10], Sang Jun Yoon[11], Sung Won Kwon [11], Jong-Lyul Park[12], In Sun Chu [12,13], Jin Man Kim[9], Seon-Young Kim [12,13,14], Yujuan Shan[15], Lihua Liu[15], Sung-A Hong[16], Dong Wook Choi[17], Junyoung O. Park [18], Young Seok Ju [3,19], Minho Shong[19], Seon-Kyu Kim [12] ✉, Bon Seok Koo [2,4] ✉ & Yea Eun Kang [1,2,20] ✉

The role of the serine/glycine metabolic pathway (SGP) has recently been demonstrated in tumors; however, the pathological relevance of the SGP in thyroid cancer remains unexplored. Here, we perform metabolomic profiling of 17 tumor-normal pairs; bulk transcriptomics of 263 normal thyroid, 348 papillary, and 21 undifferentiated thyroid cancer samples; and single-cell transcriptomes from 15 cases, showing the impact of mitochondrial one-carbon metabolism in thyroid tumors. High expression of serine hydroxymethyltransferase-2 (SHMT2) and methylenetetrahydrofolate dehydrogenase 2 (MTHFD2) is associated with low thyroid differentiation scores and poor clinical features. A subpopulation of tumor cells with high mitochondrial one-carbon pathway activity is observed in the single-cell dataset. SHMT2 inhibition significantly compromises mitochondrial respiration and decreases cell proliferation and tumor size in vitro and in vivo. Collectively, our results highlight the importance of the mitochondrial one-carbon pathway in undifferentiated thyroid cancer and suggest that SHMT2 is a potent therapeutic target.

Thyroid cancer is the most common malignancy of the endocrine system and its incidence has been rapidly increasing over the past few decades[1]. Thyroid cancer-derived follicular epithelial cells can be divided into well-differentiated thyroid cancers (DTC), including papillary thyroid cancer (PTC) and follicular thyroid cancer (FTC); poorly differentiated thyroid cancer (PDTC); and anaplastic thyroid cancer (ATC); according to their degree of differentiation[2]. DTC is mostly treated with surgery and/or radioactive iodine (RAI) and typically has a good prognosis[3–5]. However, RAI refractoriness and distant metastases may develop as the degree of differentiation decreases[3]. The average lifespan of patients diagnosed with PDTC/ATC is very short, approximately six months[6,7]. Although targeted agents such as lenvatinib and sorafenib have been approved for the treatment of thyroid cancer, their efficacy is not sufficient[8–10]. Therefore, new treatment options are required for advanced, differentiated, and undifferentiated thyroid cancers.

Cellular plasticity allows cancer cells to shift from differentiated to undifferentiated tumor[11,12]. To quantitatively measure the degree of

differentiation, the thyroid differentiation score (TDS), which scores the expression levels of genes related to iodine metabolism, was defined in The Cancer Genome Atlas (TCGA) project[8]. Several recent genomic studies have shown that the accumulation of genetic alterations, such as *TERT* and *TP53* mutations, is related to undifferentiated thyroid cancer or low TDS scores in thyroid cancer[13–17]. However, the molecular mechanisms underlying thyroid cancer dedifferentiation remain unclear and require further investigation.

Metabolic reprogramming is a common feature of tumorigenesis that exerts profound effects on gene expression, cellular differentiation, and the tumor microenvironment[18,19]. Although serine is a non-essential amino acid in humans, it has pivotal functions in cancer cells[20]. Serine and glycine are biosynthetically linked and promote rapid proliferation in tumors by providing one-carbon units for purines and NADPH production, which can support DNA/RNA synthesis during genome replication[21–23]. The serine/glycine metabolic pathway (SGP) encompasses serine/glycine uptake, the de novo serine biosynthetic pathway (SSP), and mitochondrial or cytosolic one-carbon pathways[24]. Serine for the SGP can be acquired in two ways: serine/folate uptake from the extracellular matrix and the SSP following glycolysis. The importance of the SSP is well documented in multiple cancer types[25–27]. The SSP utilizes the glycolytic intermediate 3-phosphoglycerate for serine synthesis via phosphoglycerate dehydrogenase (PHGDH), phosphoserine aminotransferase 1 (PSAT1), and phosphoserine phosphatase (PSPH)[28]. Subsequently, serine hydroxymethyltransferase (SHMT), which is encoded by cytosolic *SHMT1* and mitochondrial *SHMT2*, initiates serine catabolism[29].

SHMTs catalyze the reversible conversion of serine to glycine, which is further oxidized by the downstream 5,10-methylene-tetrahydrofolate dehydrogenase (MTHFD) enzymes, which include cytosolic MTHFD1 and mitochondrial MTHFD2, to produce NAD(P)H and formate[29]. Recently, SHMT-dependent serine catabolism was demonstrated to be the main source of one-carbon units and was critical for maintaining cellular redox control under stress conditions[30]. Moreover, *SHMT2* is critical for mitochondrial respiration and oxidative phosphorylation by stabilizing mitochondrial translation[31]. Transcriptional changes in *SHMT1* and *SHMT2* have been found in several tumors[30,32,33], and most cancer cells favor the activity of mitochondrial SHMT2 over cytosolic SHMT1 in nucleotide synthesis, suggesting an impact of mitochondrial folate synthesis on tumorigenesis[33]. In addition, binding of the 5′ untranslated region (UTR) of the *SHMT2* transcript to SHMT1 protein has been shown to inhibit the enzymatic activity of SHMT1[34], suggesting an important link between the mitochondrial and cytosolic one-carbon pathway in cancer progression. Alterations in serine levels and upregulation of SHMT2 have been identified in papillary thyroid cancer[35]. However, no previous study has dissected the role of the SGP in thyroid cancer dedifferentiation.

In this study, we investigated the crosstalk between cancer cell dedifferentiation and SGP in thyroid cancer using metabolomics, bulk RNA sequencing, and single-cell RNA sequencing data. We also validated these findings using in vitro and in vivo experiments. We demonstrated that SHMT2 could be a crucial target for the treatment of undifferentiated thyroid cancer.

## Results
### Serine/glycine metabolic pathway is critical in undifferentiated thyroid cancer
Here, we aimed to identify the key metabolic signatures of thyroid cancers. We quantified tumor-specific changes in metabolite levels in 17 thyroid cancer-normal pairs using liquid chromatography-mass spectrometry (LC-MS). We performed bulk RNA sequencing of 369 primary thyroid cancers (348 PTCs, 5 PDTCs, and 16 ATCs) and 263 adjacent normal tissues to investigate changes in the tumor-specific expression and intertumoral heterogeneity. We also performed single-

cell RNA sequencing from 15 tissues (7 PTCs, 5 ATCs, and 3 adjacent normal tissues) (Fig. 1a).

Among the measured 216 metabolites, 51 metabolites were significantly elevated while 10 metabolites were significantly decreased in tumor samples from PTC patients compared to adjacent normal tissues (Supplementary Data 1 and 2, Fig. 1b). Essential amino acids, including L-leucine, L-methionine, tryptophan, L-phenylalanine, L-valine, and L-glycine, and non-essential amino acids, L-asparagine, and L-serine, and L-alanine were significantly increased in 17 PTC tumor tissues compared to paired normal tissues (Fig. 1b). In addition, the metabolites of purine metabolism (e.g., AMP, GMP, IMP, $NAD^+$, $NADP^+$, guanine, and hypo-xanthine) were significantly increased in the tumors suggesting an impact of amino acid metabolism on thyroid cancer (Fig. 1b). Next, we performed an unsupervised analysis of transcriptomics data according to tumor type (Supplementary Fig. 1a–f and Supplementary Data 3–8). To investigate the gene signature associated with metabolic pathways in thyroid cancer, we analyzed gene expression using 369 primary tumor samples in relation to the pathologic subtypes. Gene-set enrichment analysis (GSEA) was performed to identify enriched pathways in each subtype. Multiple signaling pathways, including cell cycle and pyrimidine metabolism, were enriched in ATC compared to those in PTC (Supplementary Fig. 1d). In contrast, other metabolic pathways, such as butanoate, valine, and leucine metabolism, were enriched in PTC, confirming their metabolic differences.

Next, we assessed the thyroid differentiation score (TDS) of each tumor using the expression profile of thyroid differentiation markers and the gene set variation analysis (GSVA) algorithm[8] (Fig. 1c and Supplementary Fig. 1g). Using principal component analysis, we found that PTC and ATC showed different expression patterns, and some PTC patients showed similar gene expression to that of ATC. The distinction between high and low TDS scores was more apparent than the categorization of the tumors based on histologic types (Fig. 1c). We performed an analysis on differentially expressed genes (DEGs) and Gene Set Enrichment Analysis (GSEA) to identify significant metabolic genes that contribute to the differences between normal and tumor cells, as well as low and high TDS tumors (Fig. 1d–g and Supplementary Fig. 1g). Importantly, glycine, serine and threonine metabolism pathways were enriched in both comparisons. Of the 31 Kyoto Encyclopedia of Genes and Genomes (KEGG) pathways that showed significant differences (*p*-value < 0.05 and | Normalized Enrichment score (NES) | > 1.5) between normal and thyroid cancer tissues, six metabolic pathways had a total of 149 corresponding genes (Fig. 1f and Supplementary Data 9-10). Moreover, 371 genes in 11 metabolic pathways were identified in the TDS group comparison (Fig. 1g and Supplementary Data 11 and 12). We found that one-carbon metabolism genes, such as *SHMT2*, *TYMS*, and *MTHFD1L*, were highly ranked with significant adjusted *p*-value and fold changes in the analyses (Fig. 1f, g and Supplementary Fig. 1g). Collectively, these multi-omics data indicate the importance of SGP in thyroid cancer dedifferentiation.

### *SHMT2* and *MTHFD2* are significantly upregulated and associated with poor prognosis in undifferentiated thyroid cancer
We analyzed gene expression in 369 primary tumor samples (348 PTCs, 5 PDs, 16 ATCs; Fig. 2) and analyzed the correlation between TDS and SGP genes to investigate the role of SGP in tumor differentiation. The expression levels of mitochondrial one-carbon pathway genes (e.g., *SHMT2*: r = −0.384, *p* < 0.001 and *MTHFD2*: r = −0.381, *p* < 0.001) and serine and folate transporter genes (e.g., *SLC1A5*: r = −0.542, *p* < 0.001; *SLC1A4*: r = 0.357, *p* < 0.001; and *FOLH1*: r = −0.318, *p* < 0.001) were negatively correlated with TDS (Fig. 2a, b and Supplementary Fig. 2a). However, the expression of genes related to the SSP (e.g., *PHGDH*: r = 0.237, *p* < 0.001; *PSAT1*: r = 0.106, *p* < 0.001; and *PSPH*: r = 0.404, *p* < 0.001) and the cytosolic one-carbon pathway (e.g., *SHMT1*: r = 0.443, *p* < 0.001) was positively correlated with the TDS

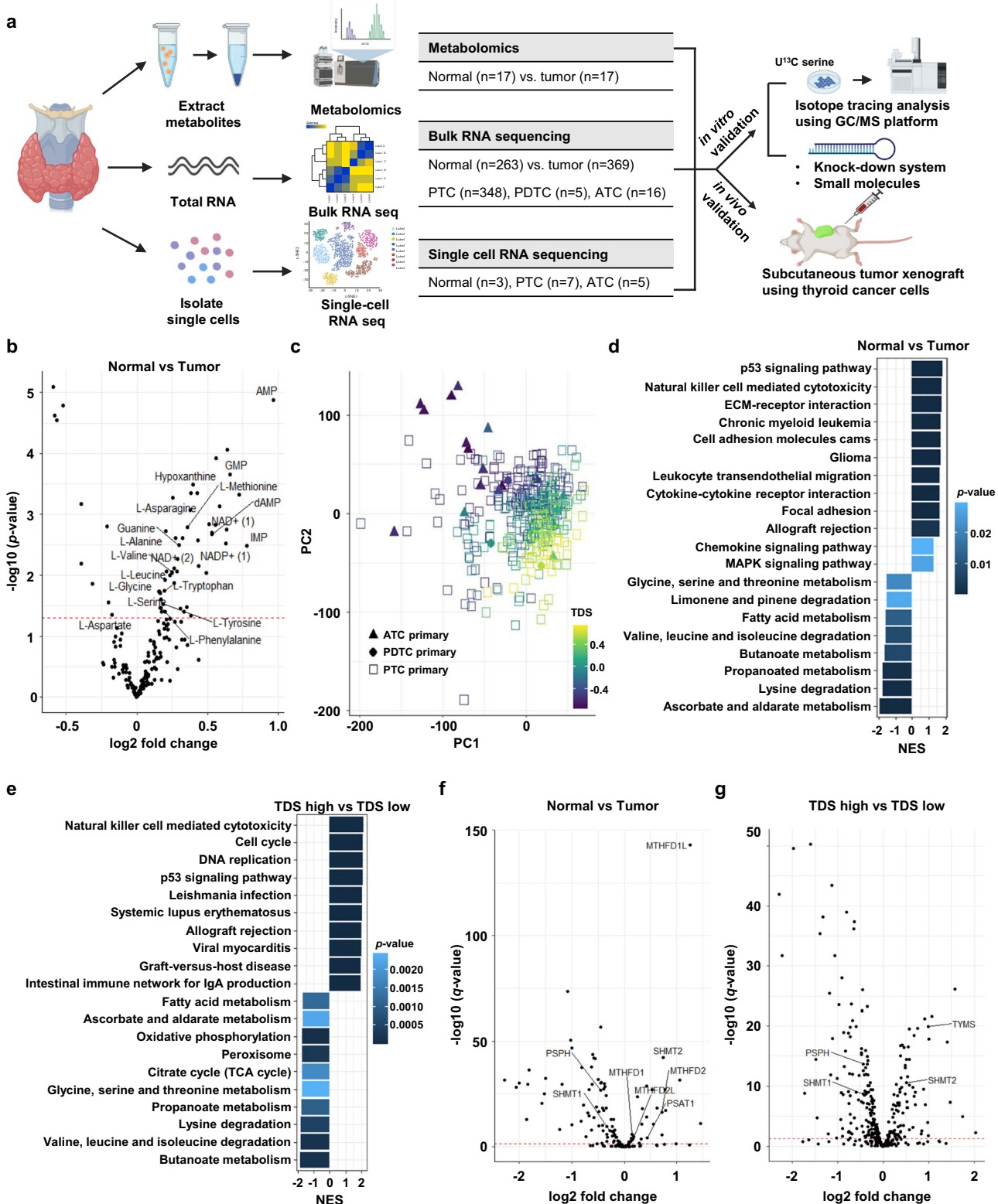

(Fig. 2a, b and Supplementary Fig. 2a). In addition, *SHMT2* (r = 0.326, *p* < 0.001) and *MTHFD2* (r = 0.241, *p* < 0.001) were positively correlated with tumor size (Fig. 2c), whereas most of the other genes in the SGP group exhibited weak (r < 0.2) or no significant correlation (*p* > 0.05). Consistently, both *SHMT2* and *MTHFD2* showed a significant negative correlation with TDS in TCGA cohort (Supplementary Fig. 2b). These results revealed the intertumoral heterogeneity of the SGP and its association with differentiation status. Undifferentiated tumors, which

are known to be more aggressive, show increased expression of mitochondrial one-carbon pathway genes. Next, we divided tumors into two groups based on the TDS scores to characterize the SGP expression pattern: TDS-high (TDS > 0) and TDS-low (TDS < 0) (Fig. 2d and Supplementary Fig. 2c). The differences in expression between the groups were well consistent with the results of correlation analyses.

Since *SHMT2* and *MTHFD2* were consistently upregulated in undifferentiated thyroid cancer and strongly positively correlated with

**Fig. 1 | The serine/glycine metabolic pathway was associated with undifferentiated thyroid cancer. a** Schematic illustration describing the design of our study to verify the role of serine/glycine metabolic pathway (SGP) in thyroid cancer (created with biorender.com). **b** Volcano plot of differentially expressed metabolites in 17 tumor-normal pairs. A student's *t* test (paired, two-sided) was used for statistical analysis. **c** Scatter plot of PC1 and PC2 from principal-component analysis (PCA) using gene expression of 369 primary tumor samples (348 PTCs, 5 PDTCs, and 16 ATCs). The shape and color of points indicate cancer type (▲, ATC; ●, PDTC; □, PTC) and TDS score, respectively. **d** Significantly enriched KEGG pathway from GSEA between tumor and normal tissues. The horizontal axis and color represent normalized enrichment score (NES) and *p*-value, respectively.

**e** Significantly enriched KEGG pathway from GSEA between TDS-high and -low tumors. The horizontal axis and color represent the normalized enrichment score (NES) and *p*-value, respectively. **f** Volcano plot comparing expression in 369 primary tumors (348 PTCs, 5 PDTCs, and 16 ATCs) and 263 normal tissues using genes from the top metabolic pathways. Names of SGP genes are annotated. NES and *p*-value was calculated using fgsea R package. **g** Volcano plot comparing expression of TDS-high ($n = 138$) and -low ($n = 231$) tumors using genes from the top metabolic pathways. Names of SGP genes are annotated. PTC papillary thyroid cancer, PDTC poorly differentiated thyroid cancer, ATC anaplastic thyroid cancer, TDS thyroid differentiation score. Source data are provided as a Source Data file.

tumor size, we evaluated the associations of these genes with clinico-pathological features. In our cohort, PDTC and ATC were significantly enriched in the *SHMT2*- or *MTHFD2*-high groups compared to the corresponding low groups (Fig. 2e and Supplementary Table 1-2). The proportion of PDTC and ATC were also significantly higher in the TDS-low group than in the TDS-high group (Fig. 2e and Supplementary Table 3). In the groups with high *SHMT2* and *MTHFD2* expression, there were more cases of larger tumor sizes and distant metastasis (Fig. 2f, g and Supplementary Table 1-2), whereas no significant difference was observed between the TDS-high and TDS-low groups (Fig. 2f, g and Supplementary Table 3). Similarly, high expression of *SHMT2* and *MTHFD2* was significantly associated with extrathyroidal extension (ETE), lymph node metastasis, stage III or IV, and the presence of the BRAF[V600E] mutation in the TCGA cohort (Supplementary Tables 4 and 5). Collectively, we found that *SHMT2* and *MTHFD2* were significantly associated with low TDS and aggressive clinico-pathological features, supporting the importance of the mitochondrial one-carbon pathway in undifferentiated thyroid carcinoma.

## Single-cell RNA sequencing validates the relationship between the mitochondrial one-carbon pathway and thyroid cancer dedifferentiation

To ascertain the correlation between SGP and thyroid cancer differentiation, we conducted single-cell RNA sequencing of a cohort of 7 PTC and 5 ATC tissues. In addition to tumor samples, our analysis included thyroid cells obtained from 3 adjacent normal tissues from our previous single-cell RNA sequencing study[36]. Multi-step filtering methods were employed to exclude low-quality cells, multiplets, and cell cycle-biased cells, ultimately yielding a dataset comprising 88,008 cells from 15 tissues (**Methods**). By leveraging known marker genes (e.g., *TG, COL1A1, CDHS*, and *PTPRC*), we successfully assigned cell types to each cluster (Fig. 3a and Supplementary Fig. 3a). Our initial comparative analysis focused on discerning the dissimilarities in the tumor microenvironment between the PTC and ATC groups. Notably, we uncovered discrepancies in the composition of the cell types between the two groups. Specifically, ATC exhibited a higher proportion of fibroblast and macrophages, while PTC displayed a greater abundance of thyroid cancer cells (Supplementary Fig. 3b). To corroborate our findings, we subjected our dataset of 369 bulk RNA sequencing data samples to CIBERSORTx analysis, employing a signature matrix derived from our single-cell RNA sequencing results[37] (Supplementary Fig. 3c). Furthermore, we performed GSEA to identify enriched pathways specific to PTC or ATC within each cell type. Genes associated with oxidative phosphorylation and antigen presentation were significantly enriched in ATC across multiple cell types, whereas genes involved in gap junction were enriched in PTC (Supplementary Fig. 3d). The convergence of these common pathways across various cell types highlights the existence of distinct cellular networks unique to each cancer type and warrants further investigation.

Subsequently, we compared the expression of thyroid-origin cells among the groups to identify the intrinsic differences in tumor cells. It has come to our attention that when utilizing anchor-based integration, there is a tendency for overcorrection of the batch in thyroid-origin cells (Supplementary Fig. 4a). To overcome this problem, we employed the ComBat-seq algorithm to perform batch-effect correction on the entire gene expression matrix of thyroid-origin cells, ensuring unbiased comparisons across all expressed genes[38]. This process made a better distinction between PTC, ATC, and normal thyroid cells (Fig. 3b). Notably, ATC demonstrated significant enrichment of genes and pathways associated with mitochondrial metabolic pathways, including oxidative phosphorylation and electron-transport chain (Supplementary Fig. 4b).

Our focus shifted to the genes associated with SGP. We observed elevated expression levels of *SHMT2* and *MTHFD2* genes in cancer cells compared to normal cells (*p*-values < 0.001; Fig.3c). Moreover, ATC cells exhibited higher expression levels of these genes compared to PTC cells (*SHMT2, p*-value < 0.01; *MTHFD2, p*-value < 0.05). Genes in the other compartments of the SGP demonstrated various expression patterns (Supplementary Fig. 4c). Of particular significance, *PHGDH* in the SSP and *SHMT1* in the cytosolic one-carbon pathway manifested low expression in ATC and high expression in normal tissues, which was in contrast of mitochondrial one-carbon pathway genes (Fig. 3c). In addition, genes related to glycolysis, such as *SLC2A3* and *GAPDH*, demonstrated elevated expression in ATC cells compared to PTC cells (Supplementary Fig. 4d). These findings were consistent with the results obtained from bulk RNA sequencing, further substantiating the pivotal role of mitochondrial one-carbon pathway enzymes in undifferentiated thyroid cancer metabolism.

Next, we examined the intricate connection between SGP genes and TDS by analyzing the average gene expression of individual cells from each tissue (Fig. 3d). These average values represented pseudo-bulk tissues solely comprising in tumor cells or normal thyroid follicular cells, uncontaminated by other cell types. Consistent with the findings from bulk RNA sequencing, mitochondrial one-carbon pathway enzymes (*SHMT2* and *MTHFD2*) were negatively correlated with TDS (*SHMT2*, r = −0.64, *p*-value = 0.01; *MTHFD2*, r = −0.68, *p*-value = 0.005; Fig. 3d). In contrast, *SHMT1* and *PHGDH* were positively correlated with TDS (*SHMT1*, r = 0.95, *p*-value < 0.001; *PHGDH*, r = 0.49, *p*-value = 0.06; Fig. 3d). These results reaffirmed the different roles of the SGP compartments in thyroid cancer dedifferentiation. Similar correlations were also observed at the single-cell level (Supplementary Fig. 5a). Despite the substantial overlap of expression between ATC and PTC, we detected a specific distribution in cells with increased TDS (TDS-high) and cells with increased expression of *SHMT2* and *MTHFD2* (SM-high; Fig. 3e, Supplementary Fig. 5b). To gain further insights into the cellular relationship between TDS-high cells and SM-high cells within tumors, we employed the RNA velocity algorithm to infer the trajectory of single cells[39]. There was a convergence towards TDS-low SM-high cells, aligned with the dedifferentiation process of thyroid cancer cells (Fig. 3f).

To comprehensively investigate the gene expression disparities between TDS-high SM-low cells and TDS-low SM-high cells, we conducted a comparative analysis and discovered many DEGs in both groups, supporting the existence of distinct cellular states (Fig. 3g). Using GSEA with Gene Ontology (GO) biological processes, we identified divergently activated pathways in each subgroup

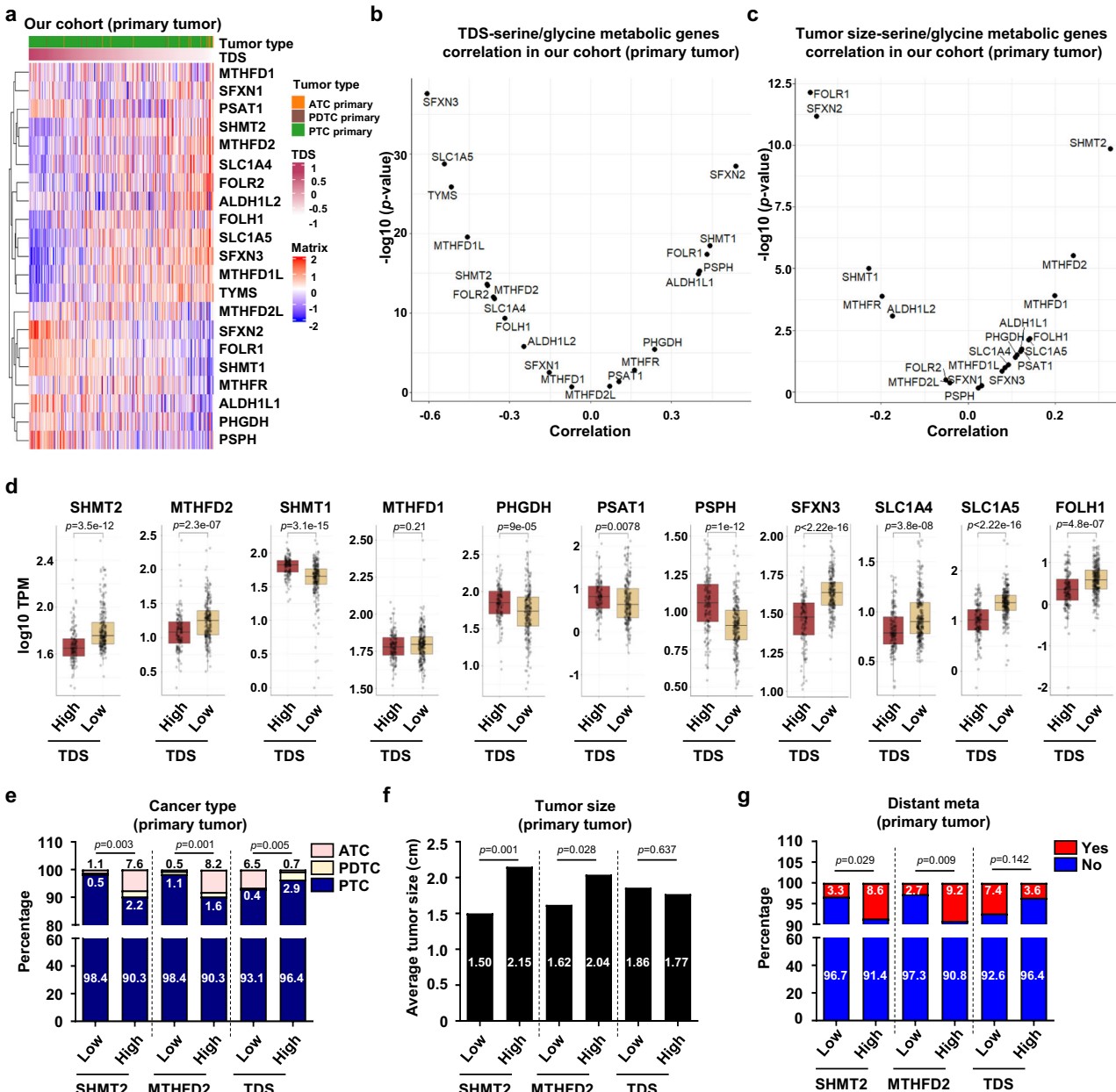

**Fig. 2 | *SHMT2* and *MTHFD2* were upregulated and associated with poor prognosis in undifferentiated thyroid cancer. a** Heatmap showing normalized expression of serine/glycine metabolic pathway (SGP) genes in 369 primary tumor samples (348 PTCs, 5 PDTCs, and 16 ATCs). Tumor type were defined by color (green, PTC; brown, PDTC; orange, ATC). TDS was defined by gradient color. **b** Scatter plot showing Pearson's correlation between thyroid differentiation score (TDS) and expression of SGP genes in 369 primary tumor samples (348 PTCs, 5 PDTCs, and 16 ATCs). **c** Scatter plot showing Pearson's Correlation between tumor size and expression of SGP genes in 369 primary tumor samples (348 PTCs, 5 PDTCs, and 16 ATCs). **d** Box plots comparing gene expression of SGP genes between TDS-high ($n = 138$) and -low ($n = 231$) tumors. The SGP genes are involved in the mitochondrial (*SHMT2* and *MTHFD2*) and cytosolic (*SHMT1* and *MTHFD1*) one carbon pathways, the SSP (*PHGDH*, *PSAT1*, and *PSPH*), mitochondrial serine transporter (*SFXN3*), cellular serine transporter (*SLC1A4* and *SLC1A5*), and folate

transporter (*FOLH1*). Data were expressed as the mean ± SD. A student's *t* test (two-sided) was used for statistical analysis. **e** Bar plots comparing the proportion of cancer types between low and high groups of *SHMT2*, *MTHFD2*, or TDS in primary tumors. Tumor type were defined by color (blue, PTC; light yellow, PDTC; light pink, ATC). A chi-square test (two-sided) was used for statistical analysis. **f** Bar plots comparing tumor size between low and high groups of *SHMT2*, *MTHFD2*, or TDS in primary tumors. A chi-square test (two-sided) was used for statistical analysis. **g** Bar plots comparing the proportion of tumors with distant metastasis between low and high groups of *SHMT2*, *MTHFD2*, or TDS in primary tumors. Blue, without distant metastasis; Red, with distant metastasis. PTC papillary thyroid cancer, PDTC poorly differentiated thyroid cancer, ATC anaplastic thyroid cancer, TPM transcripts per million, Distant meta distant metastasis. A chi-square test (two-sided) was used for statistical analysis. *, $p < 0.05$; **, $p < 0.01$; ***, $p < 0.001$; ****, $p < 0.0001$; ns, not significant. Source data are provided as a Source Data file.

(Fig. 3h and Supplementary Fig. 5c). As expected, the pathway associated with thyroid hormone generation (GO:0006590) was enriched in TDS-high SM-low cells (normalized enrichment score [NES] = 1.97, *p*-value = 0.001), whereas mitochondria-related pathways, including oxidative phosphorylation (OXPHOS, GO:0006119,

NES = 2.62, *p*-value = 0.004) and mitochondrial membrane organization (GO:0007006, NES = 2.33, *p*-value = 0.004) were enriched in TDS-low SM-high cells. These results exclude the possibility of stochastic or artifactually increasing gene expression in the subpopulation of cells and corroborate the presence of cells with

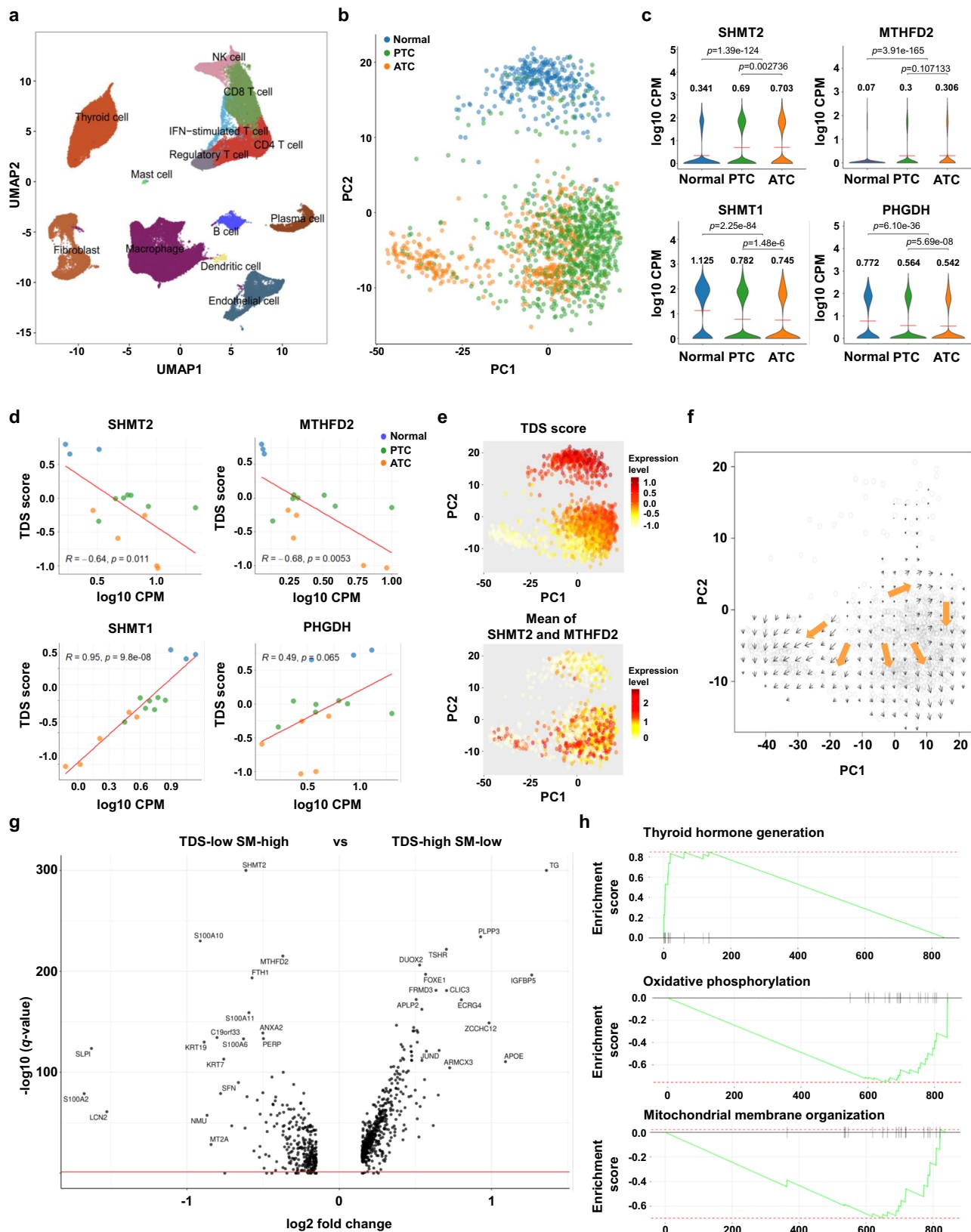

activated mitochondrial one-carbon and relevant pathways. Overall, the analysis conducted using single-cell RNA sequencing reaffirmed the inverse relationship between thyroid differentiation and the mitochondrial one-carbon pathway while revealing the existence of a distinct subpopulation of cells characterized by heightened mitochondria-related metabolism.

## Down-regulated *SHMT2* inhibits cell viability and migration in undifferentiated thyroid cancer cells

Before observing the in vitro phenotype resulting from SHMT2 suppression, we examined the protein expression of SHMT2 in the original thyroid cancer cell lines. SHMT2 was expressed in all thyroid cancer cell lines, including normal thyroid (Nthy-ori 3-1), PTC

**Fig. 3 | Single-cell RNA sequencing exhibited the association between the mitochondrial one-carbon pathway and thyroid cancer dedifferentiation.**
**a** UMAP plot of all single cells colored by 88,008 cell types. **b** PCA plot of thyroid cells colored by tissue types. Blue, normal thyroid cells; Green, PTC cells; Orange, ATC cells. In order to achieve balanced cell counts across samples, a total of 100 cells were randomly chosen for each sample. **c** Violin plots comparing the expression levels of *SHMT2, MTHFD2, SHMT1,* and *PHGDH* among different tissue types. Mean values are marked by a red line and are written at the top of each plot. A student's *t* test (two-sided) was used for statistical analysis. **d** Scatter plots showing the relationship between gene expression level (log10 CPM) and thyroid differentiation score (TDS). Each dot represents the mean value of each tissue, and the red lines indicate the fitted linear regression. Blue, normal thyroid cells; Green, PTC cells; Orange, ATC cells. A pearson's correlation coefficients test (*p*; two-side) was used for analysis. **e** PCA plots colored by TDS (upper) and the mean CPM value of *SHMT2* and *MTHFD2* (lower). **F** Result of trajectory inference by the RNA velocity algorithm. Black arrows represent the vector from the current transcriptional state to the estimated future state. Orange arrows indicate the summarized direction of regional vectors from TDS-high cells to *SHMT2/MTHFD2*-high cells. **g** Volcano plot showing differentially expressed genes between TDS-high SM-low and TDS-low SM-high cells. *p*-value was calculated using DESeq2 R package. **H** Representative enrichment plots from GSEA between TDS-high SM-low and TDS-low SM-high cells. PTC papillary thyroid cancer; ATC anaplastic thyroid cancer, CPM count per million, SM mean of *SHMT2* and *MTHFD2* expression levels; **, *p* < 0.01; ***, *p* < 0.001; ns, not significant. Source data are provided as a Source Data file.

(TPC-1 and BCPAP), and ATC (8505 C and FRO). SHMT2 and MTHFD2 were highly expressed in both PTC cell lines and ATC cell lines compared to normal cells (Supplementary Fig. 6a). There was no significant difference in the expression of SHMT2 and MTHFD2 in patient-derived PTC cell lines and ATC cell lines, which was different from our RNA sequencing results. SHMT2 and MTHFD2 protein expression were more prominently elevated in ATC than in PTC or normal tissues, similar to the results from the RNA sequencing data (Supplementary Fig. 6b).

To investigate the effect of SHMT2 suppression in thyroid cancer cells, transient knockdown of SHMT2 was performed in PTC (BCPAP) and ATC (8505 C and FRO) cell lines (Supplementary Fig. 6c–h). *SHMT2* knockdown significantly reduced the proliferation of 8505 C and FRO cells, and the results of BCPAP were not clear as they differed depending on the type of sequence of si-RNA. (Supplementary Fig. 6d, f, and h). To confirm the effect of *SHMT2* on thyroid cancer, *SHMT2* was knocked down using shRNA lentiviral particles (Fig. 4 and Supplementary Fig. 7). The stable knockdown of *SHMT2* reduced the proliferation of 8505 C cells (Fig. 4a). Cell migration was also reduced by approximately 50% in shSHMT2-8505C cells compared to shControl-8505C cells (Fig. 4b). In shSHMT2-TPC1 cells, there was significantly reduced cell viability, migration, and oxygen consumption compared to shControl-TPC1 cells (Supplementary Fig. 7e–h) and shSHMT2-BCPAP cells showed significantly reduced cell viability, and migration was not changed compared to shControl-BCPAP cells (Supplementary Fig. 7j, k).

As serine catabolism by SHMT2 is required for efficient cellular respiration and the assembly of complex I in the respiratory chain[31,40], we analyzed the effects of SHMT2 on mitochondrial respiration. The shSHMT2-8505C cells showed a significantly reduced oxygen consumption rate (OCR) including basal and maximal respiration (Fig. 4c). Similar to the results for shSHMT2-8505C cells, cell viability, migration, and mitochondrial function were reduced by more than 50% in shSHMT2-FRO-luc cells compared to shControl-FRO-luc cells (Fig. 4d–f). Moreover, the expression of OXPHOS complex proteins was reduced by the knock-down of *SHMT2* (Fig. 4g). We confirmed the growth of shSHMT2 thyroid cancer cell lines using WST-1 and apoptosis assays, supplemented with and without formate, to test for rescue. Importantly, the effects of reduced SHMT2 on cell viability and apoptotic cell death could be rescued by formate, indicating that SHMT2 inhibits cell viability through the on-target depletion of formate (Fig. 4h–j).

Next, to identify the role of SHMT2 while excluding off-target effects, we used the CRISPR/Cas9 system in thyroid cancer cell lines to confirm the effect of *SHMT2* downregulation. *SHMT2* expression was significantly reduced by a single-guide RNA (sgRNA) sequence targeting *SHMT2* using the CRISPR/Cas9 system in the PTC (BCPAP) and ATC (8505 C and FRO) cell lines. Cell viability and migration were effectively reduced in sgSHMT2-cell lines compared to controls (Supplementary Fig. 8-9). Furthermore, the effect of reduced *SHMT2* expression on cell viability was rescued by the addition of formate to

8505 C and FRO cells (Supplementary Fig. 9g, h). These results suggest the importance of the mitochondrial folate cycle in thyroid cancer and that the inhibition of SHMT2 can effectively regulate tumor aggressiveness in thyroid cancer.

## SHMT2 inhibition reduced cell growth and mitochondrial function in undifferentiated thyroid cancer

The functional importance of SHMT2 in thyroid cancers has led to the examination of metabolic pathways associated with the folate cycle in vitro. We investigated the baseline metabolic activity of SHMT2 and the effect of SHMT2 inhibition in thyroid cancer cells using SHIN2, an SHMT inhibitor that was previously developed as a small molecule to attenuate tumor growth in leukemia[41]. To evaluate SHIN2 target engagement in undifferentiated thyroid cancer, we followed a previous assay based on the continuous infusion of tracer amounts of [U-$^{13}$C$_3$]-serine (which contains three $^{13}$C atoms and is accordingly M + 3) and measured serine and glycine labeling by mass spectrometry (Supplementary Fig. 10a). When metabolized through SHMT activity, M + 3 serine is converted to M + 2 glycine and M + 1 methylene-THF. Because of the reversibility of SHMT activity, M + 2 glycine can recombine with an unlabeled one-carbon unit from methylene-THF to give rise to M + 2 serine, and unlabeled glycine can recombine with a labeled one-carbon unit to give rise to M + 1 serine (Supplementary Fig. 10a). Thus, the proportion of M + 1 or M + 2 serines indicates SHMT activity. M + 1 and M + 2 serine were significantly upregulated in ATC cells (8505 C and FRO) compared to those in PTC cells (BCPAP) at baseline (Fig. 5a), suggesting a role for SHMT2 in undifferentiated thyroid cancer. SHIN2 treatment reduced M + 2 glycine, M + 1 serine, and M + 2 serine levels in all thyroid cancer cell lines. Moreover, SHIN2 inhibition changed the metabolites in ATC cell lines to levels similar to that of PTC, although the baseline level was higher than that of the PTC cell line (Fig. 5a).

Next, we examined the effects of SHMT2 inhibition using SHIN2 on thyroid cancer cells. SHIN2 treatment induced a dose-dependent decrease in the viability of both PTC (TPC-1 and BCPAP) and ATC (8505 C and FRO) cells (Supplementary Fig. 10b–e). We examined the contribution of formate, an alternative carbon source, in the role of SHMT2 in thyroid cancer cell lines. The effect of SHIN2 on cell viability was rescued by formate addition (Fig. 5b–d). Additionally, apoptotic cell death induced by SHIN2 (20 μM, 24 h) was also rescued by supplementing formate (1 mM, 24 h; Fig. 5e, f and Supplementary Fig. 11a, b), suggesting the impact of SGP pathways through SHMT activity in thyroid cancer.

Similar to the *SHMT2* knock-down results, both 8505 C and FRO cells showed significantly less mitochondrial respiration following SHIN2 treatment (20 μM, 24 h) (Fig. 5g, h and Supplementary Fig. 11c, d), and OXPHOS complex-related protein expression (complexes I-V) was decreased in the 8505 C and FRO cells (Fig. 5I, j). These results suggest that serine catabolism by SHMT2 is required for biogenesis of the respiratory chain complex and efficient cellular respiration in undifferentiated thyroid cancer cells.

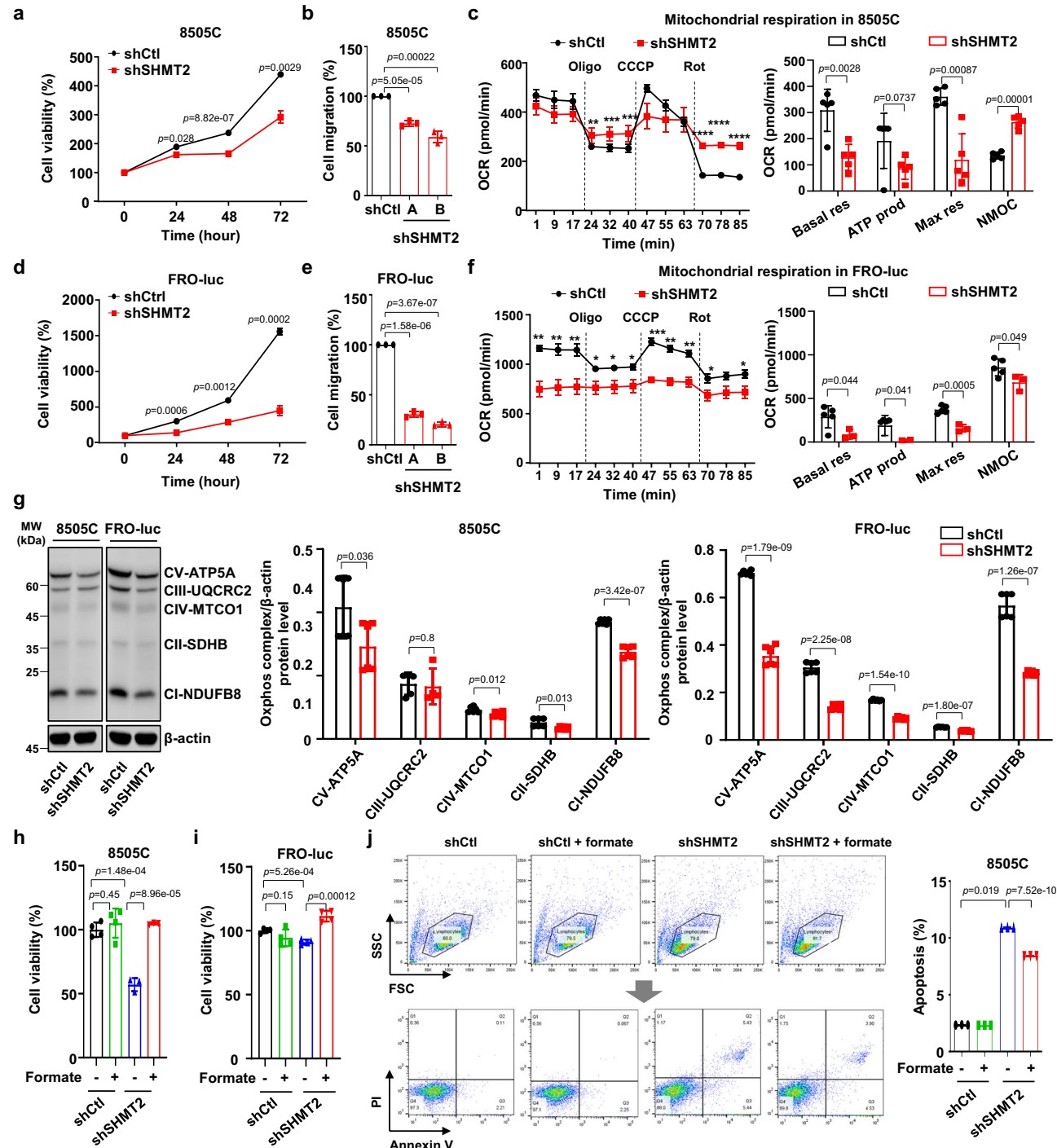

**SHMT2 inhibition reduced tumor growth in a xenograft model of undifferentiated thyroid cancer**

Finally, we verified whether SHMT2 knockdown or inhibition reduced tumor growth in vivo. To investigate these effects, shSHMT2-FRO-luc and shControl-FRO-luc cells were administered to nude mice (Fig. 6a), as described in the Methods section. We identified a reduction in SHMT2 expression by immunohistochemical staining in shSHMT2-FRO-luc-injected mice (Supplementary Fig. 12). A significant decrease in tumor mass was induced by the stable knockdown of *SHMT2* compared to the control, which was measured in the region of interest (ROI) using bioluminescence imaging (Fig. 6a). Tumor volumes were calculated using the following equation: length × width[2] × 0.5. The estimated tumor volume in shSHMT2-FRO-luc-injected mice was significantly lower than that in shControl-FRO-luc-injected mice (Fig. 6b).

Furthermore, SHIN2 treatment induced a reduction in tumor growth (Fig. 6c) compared to control mice. Taken together, our results demonstrate the essential role of SHMT2 in the tumorigenesis of undifferentiated thyroid cancer.

## Discussion

As thyroid tumors progress and tumor cells lose their ability to take up iodide, thyroid cancer becomes resistant to traditional therapeutic strategies, and the prognosis worsens[42]. Despite having the same origin as normal thyroid follicular cells, PTC, PDTC, and ATC represent heterogeneous pathologies, biological behaviors, and clinical courses. Many previous studies have found that certain molecules and pathways are involved in cancer dedifferentiation. TGF-β, Wnt, and Hedgehog signals have been found to correlate with dedifferentiation

**Fig. 4 | Down-regulated *SHMT2* reduced cell viability and migration in undifferentiated thyroid cancer cells. a** Line plot showing cell viability of shSHMT2-8505C cells compared to shControl-8505C cells in a time-dependent manner. Error bars indicate standard deviations from *n* = 3 biological replicates. **b** Bar plot showing reduced cell migration of shSHMT2-8505C cells compared to shControl-8505C cells. Error bars indicate standard deviations from *n* = 3 biological replicates. **c** Line and bar plots comparing oxygen consumption rate (OCR) of mitochondrial respiration between shSHMT2-8505C and shControl-8505C cells. Basal respiration (Basal res), ATP production (ATP prod), maximal respiration (Max res), and non-mitochondrial respiration (NMOC) were used as respiratory parameters. Data were expressed as the mean ± SD from *n* = 5 biological replicates. **d** Line plot showing cell viability measured during 72 h in shControl-FRO-luc and shSHMT2-FRO-luc cells. Error bars indicate standard deviations from *n* = 3 biological replicates. **e** Bar plot comparing cell migration between shControl-FRO-luc and shSHMT2-FRO-luc cells. Error bars indicate standard deviations from *n* = 3 biological replicates. **f** Line and bar plots comparing OCR of mitochondrial respiration between shSHMT2-FRO-luc and shControl-FRO-luc cells. Exact *p* values can be found in the source data file.

Data were expressed as the mean ± SD from *n* = 3 independent experiments. **g** Western blot images and bar plots comparing OXPHOS protein expression in *SHMT2* knock-down 8505 C and FRO-luc cells compared to control cell lines. Error bars indicate standard deviations from *n* = 4 independent experiments. **h** Bar plot comparing cell viability of shSHMT2-8505C and shControl-8505C cells with or without 1 mM formate for 24 h. Data were expressed as the mean ± SD from four independent experiments. **i** Bar plot showing reduced cell viability following SHMT2 reduction in FRO-luc cells and rescue by formate addition. Error bars indicate the standard deviation from four independent experiments. **j** Scatter and bar plots showing increased apoptotic cell death in shSHMT2-8505C cells with or without 1 mM formate compared to shControl-8505C cells. These experiments are representative of three independent experiments. Data were expressed as the mean ± SD. Oligo oligomycin; CCCP; Rot rotenone; Basal res basal respiration; ATP prod ATP production; Max res maximal respiration; NMOC non-mitochondrial respiration; FRO-luc luciferase-expressing FRO; PI propidium iodide. A student's *t* test (two-sided) was used for statistical analysis. *, $p < 0.05$; **, $p < 0.01$; ***, $p < 0.001$; ****, p < 0.0001. Source data are provided as a Source Data file.

at the invasion front of colorectal cancer[43], and dedifferentiation of glioblastoma is known to be related to chemoresistance through EGF signaling under hypoxic conditions[44]. Recently, a potential driving role of CREB3L1 in dedifferentiation was suggested based on single-cell RNA sequencing in thyroid cancer[45], and aberrant metabolic genes were shown to be a signature of dedifferentiation by bulk RNA sequencing in large cohorts[46]. Although previous studies have elucidated the role of these molecular processes, research on the mechanism of dedifferentiation in thyroid cancer, especially regarding SGP, is still lacking.

Serine has recently attracted attention as a key metabolite in the development, progression, and maintenance of cancer cells[47–50]. Several studies have revealed the importance of the SSP in differentiated thyroid cancer using immunohistochemical analysis and the expression level of PHGDH, the rate-limiting enzyme in the SSP[51,52]. The source of serine includes uptake from the extracellular matrix via serine and folate transporters, as well as the glycolysis pathway[47,50]. Some tumors show increased serine or folate uptake, and others represent the highly activated SSP[53]. Our data revealed an increase in serine and related metabolites, along with an increased expression of serine/folate transporters in undifferentiated cancer, but there was no increase in expression of SSP-related genes, such as *PHGDH*, *PSPH*, and *PSAT1*, compared to differentiated cancer. This implies that the sources of serine vary according to cancer types and are tightly associated with other cellular phenotypes, such as differentiation status.

We found that mitochondrial *SHMT2* and *MTHFD2* negatively correlated with TDS, and that both genes were consistently related to tumor size and other aggressive clinico-pathological factors. SHMT2 is known to release one-carbon units through the conversion of serine into glycine and could affect the robust energy synthesis in cancer cells and the production of NADPH to support the redox balance under hypoxic conditions[54,55]. Numerous studies have demonstrated the importance of the mitochondrial one-carbon pathway in tumorigenesis and drug resistance[56,57]. In *MYC*-amplified neuroblastoma, hypoxic stress induces the expression of SHMT2, and knock-down of SHMT2 reduces tumorigenesis[55]. In breast cancer, the mitochondrial one-carbon pathway is upregulated in metastatic subclones, and SHMT2 inhibition impairs the growth of lung metastases[58]. The inhibition of SHMT1 and SHMT2 significantly reduces tumorigenesis and is synergistically effective with methotrexate treatment in T-cell acute lymphoblastic leukemia[41,59]. Therefore, metabolic reprogramming via the upregulation of SHMT2 provides advantages for cancer cells under hypoxia or metastatic conditions. However, the role of SHMT2 in the process of dedifferentiation is not currently understood.

Recent studies have reported that serine and genes in the SGP (*PHGDH* and *SHMT2*) are significantly increased in aggressive PTC,

particularly in the presence of the BRAF^V600E mutation[35,51,52]. In addition, NCT-503, a PHGDH inhibitor, has been reported to reduce cell proliferation and tumor growth in thyroid cancer[52]. However, the effects of SHMT2 inhibition on thyroid cancer cells have not yet been elucidated. Our study directly demonstrated the therapeutic potential of targeting SHMT2 in undifferentiated thyroid cancer using both a genetic and pharmaceutical approach. Both the genetic knock-down of *SHMT2* and treatment with an SHMT inhibitor inhibited the proliferation and migration of undifferentiated thyroid cancer cells and significantly reduced tumor growth in ATC in vivo. In our experiments using thyroid cancer cells lines, knockdown or inhibition of SHMT2 showed similar effects in cell lines derived from PTC or ATC. Heterogeneity in SHMT2 levels in human ATC/PTC cell lines, differently from what we observe in human primary samples, might explain these results.

Electron transport chain dysfunction due to mitochondrial DNA (mtDNA) depletion was recently shown to dramatically alter the expression of SHMT2 and the production of one-carbon units from serine catabolism[60,61]. Other studies have shown that serine catabolism by SHMT2 is required to maintain mitochondrial respiration and translation by providing NADH and formylmethionyl-tRNAs[31,40,62], suggesting an association between mitochondrial serine catabolism and the modulation of the OXPHOS system. We identified an association of *SHMT2* expression with mitochondrial respiration and OXPHOS through pathway analysis of single-cell RNA sequencing data and inhibition of SHMT in ATC cell lines. Our observations strongly suggest a functional coupling between the SGP, especially the mitochondrial one-carbon pathway, and the OXPHOS system during dedifferentiation.

In conclusion, we identified a crucial role of mitochondrial SHMT2 in undifferentiated thyroid cancer using a large-scale multi-omics database, including metabolomics and bulk and single-cell RNA sequencing data. We validated the results using $^{13}$C serine tracing, gene knock-down in cell lines, and gene inhibition in a tumor xenograft model. Our findings highlight the importance of the mitochondrial one-carbon pathway, including the SGP, in undifferentiated thyroid cancer, and suggest SHMT2 as a potent therapeutic target.

# Methods
## Study population
Tissues were obtained from 632 patients in this study, including 263 normal thyroid tissues and thyroid tissues from 348 patients with PTC, five with PDTC, and 16 with ATC. All the samples were treatment-naive primary tumors that were obtained by surgical resection. This study was approved by the Institutional Research and Ethics Committees of the Chungnam National University Hospital (CNUH-2022-11-004-001), Seoul National University Hospital

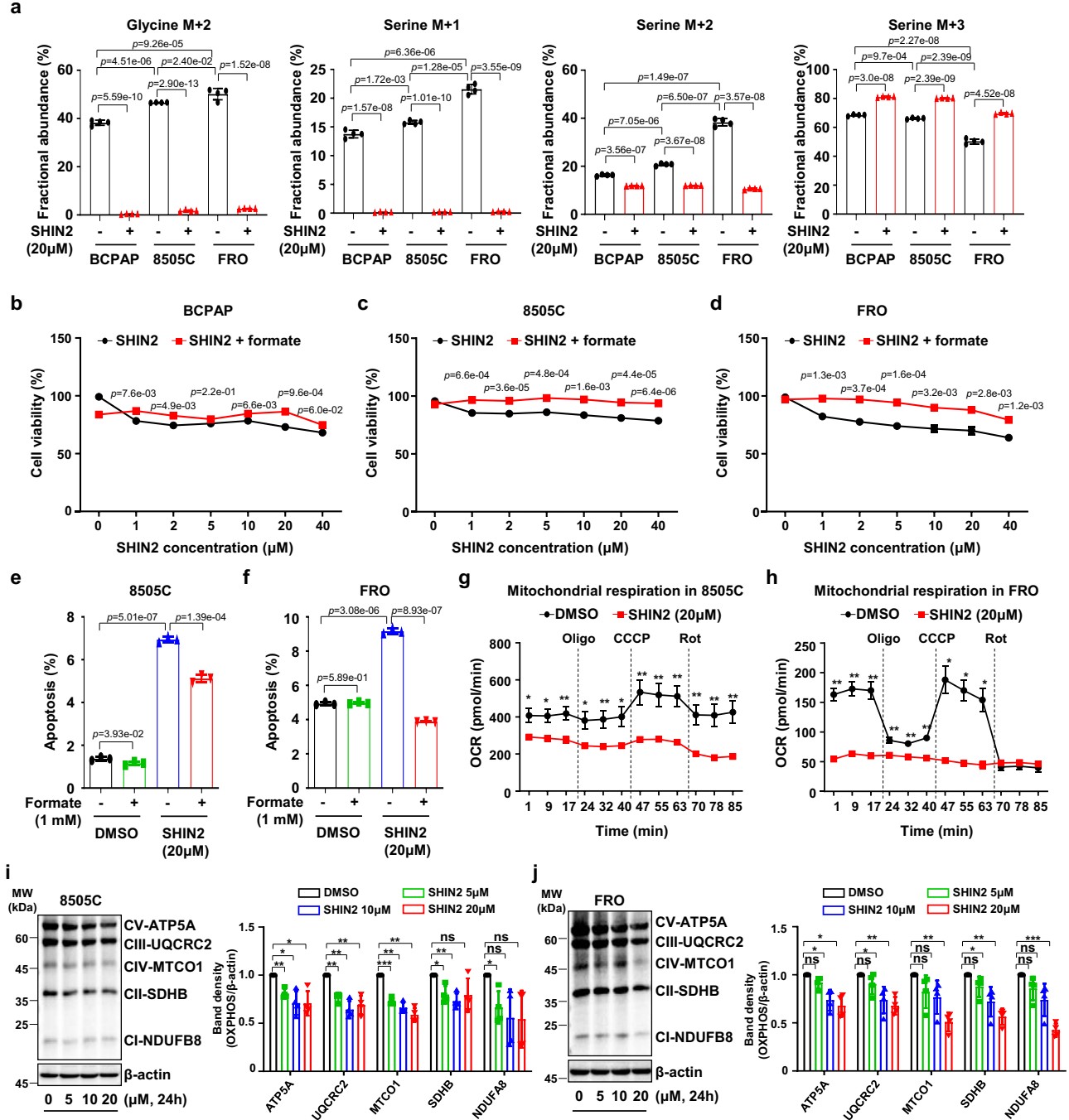

**Fig. 5 | SHIN2 reduced cell growth and mitochondrial function via SHMT inhibition in vitro. a** Bar plots showing fractional abundance of glycine m + 2, serine m + 1, serine m + 2, and serine m + 3 ($n = 4$ per group from three independent experiments). **b**–**d** Line plots showing cell viability after SHIN2 treatment with or without 1 mM formate for 24 h in BCPAP, 8505 C, and FRO cells. Error bars indicate standard deviations from three independent experiments. Black circle, SHIN2; Red square, SHIN2 with formate. **e**, **f** Bar plots showing increased apoptosis after 20 μM SHIN2 treatment for 24 h in 8505 C and FRO cells. Error bars indicate standard deviations from $n = 3$ biological replicates. **g**, **h** Line plots showing OCR of mitochondrial respiration after 20 μM SHIN2 treatment for 24 h in 8505 C (**g**) and FRO

(**h**) cells. These experiments are representative of three independent experiments. Black circle, SHIN2; Red square, SHIN2 with formate. **i**, **j** Western blot images and bar plots showing total OXPHOS protein expression after 5–20 μM SHIN2 treatment in 8505 C (**i**) and FRO (**j**) cells. Black, DMSO; Green, SHIN2 5 μM; Blue, SHIN2 10 μM; Red, SHIN2 20 μM; OXPHOS, oxidative phosphorylation. All experiments were performed independently at least three times. Data were expressed as the mean ± SD. A student's *t* test (two-sided) was used for statistical analysis. *, $p < 0.05$; **, $p < 0.01$; ***, $p < 0.001$; ns, not significant. Exact $p$ values shown in g-j can be found in source data file. Source data are provided as a Source Data file.

(H-1508-147-700) and Seoul National University Bundang Hospital (B-2012/657-308). Informed consent was obtained from all the participants. All participants gave consent to provide gender and age in accordance with the IRB, and the study was conducted

without compensation. In this study, maximum tumor size was not utilized for enrollment criteria. In the case of undifferentiated cancer, debulking surgery or total thyroidectomy is the principle of treatment regardless of the size of the tumor.

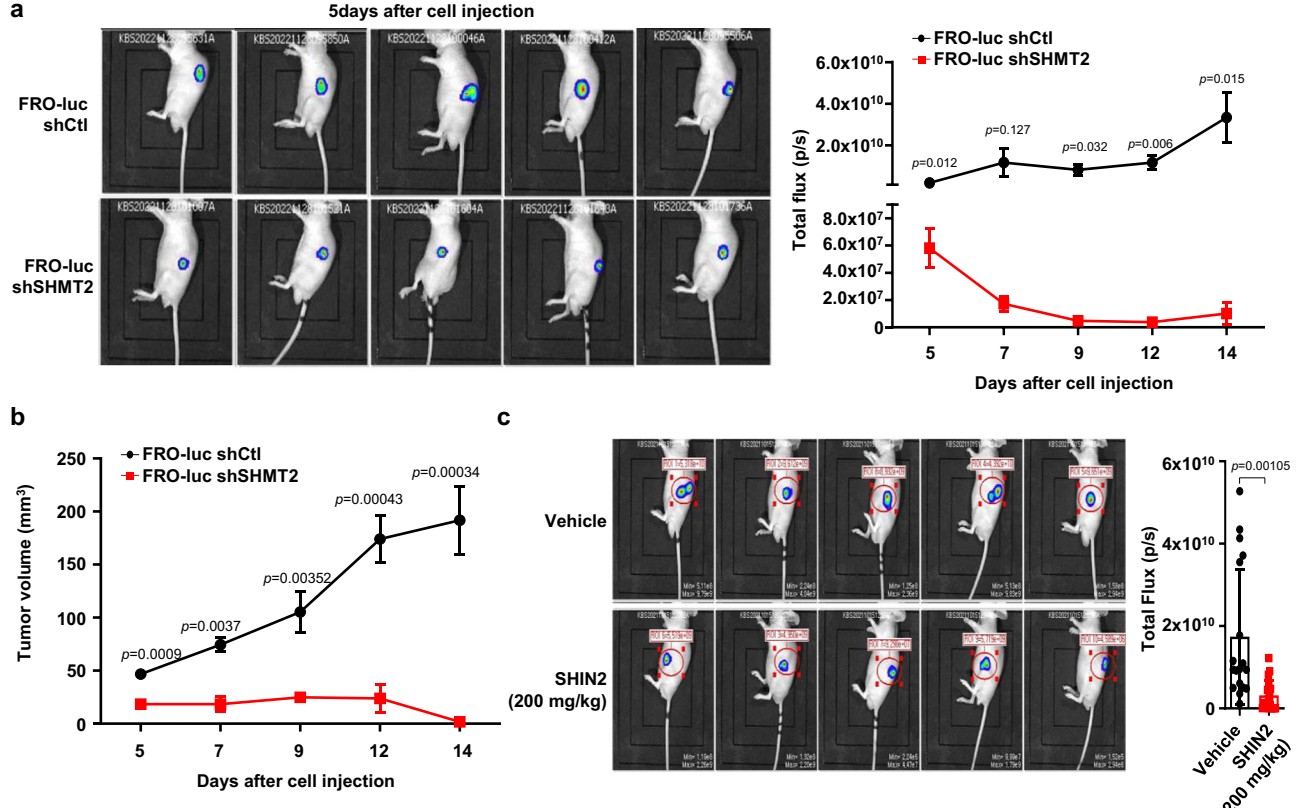

**Fig. 6 | Inhibition of SHMT2 reduced tumor growth in a xenograft model of undifferentiated thyroid cancer. a** Bioluminescence images and line plot showing tumor volume after injection of shControl/shSHMT2-FRO-luc cells into mice. For in vivo experiments, $n = 10$ mice per group from three independent experiments. **b** Line plot showing tumor volume every other day for two weeks after shControl (black circle)/shSHMT2-FRO-luc cells (red square) injection ($n = 5$ mice per group from three independent experiments). **c** Bioluminescence images and bar plot showing tumor volume in BALB/c nude mice injected with FRO-luc cells treated with/without SHIN2 ($n = 18$ mice per group from three independent experiments). These experiments are representative of three independent experiments. Data were expressed as the mean ± SD. A student's t test (two-sided) was used for statistical analysis. Source data are provided as a Source Data file.

## HPLC-QqQ MS-based targeted metabolomics

For metabolite extraction, 80% methanol (−80 °C)(1.5 mL) was added to the Precellys lysis kit including each tissue sample (10−20 mg). After homogenization with bead beaters, the tissue lysates were incubated in a thermoshaker (1200 rpm, 1 h, 4 °C). Centrifugation at 16,000 ×g for 10 min at 4 °C was followed by incubation, and supernatant (1 mL) was purged by nitrogen gas to obtain the metabolite extracts. The fully dried extracts were kept at −80 °C until instrumental analysis. The extracts were reconstituted with water (50 µL) immediately before analysis. For data acquisition, an Agilent 1260 HPLC system with an Agilent 6460 triple quadrupole (QqQ)-based mass spectrometer (MS) was used, and data extraction was performed using XCMS Online (https://xcmsonline.scripps.edu). For data processing and statistics, t-tests were used after sum normalization, log transformation, and Pareto scaling using MetaboAnalyst (https://www.metaboanalyst.ca). The detailed parameters for data acquisition and processing are described in a previous study[63].

## RNA sequencing

RNA was extracted from tumor and paired non-tumor tissue samples obtained from patients with thyroid cancer (Supplementary Data 13). Most RNA was extracted from fresh frozen tissues ($n = 552$), but some tissues were obtained from FFPE ($n = 80$). Total RNA was extracted using an RNA extraction kit (Qiagen) following the manufacturer's protocol. The quality of extracted RNA was evaluated using an Agilent 2100 Bioanalyzer RNA Nano Chip (Agilent Technology Inc., Santa Clara, CA, USA). The extracted RNA was used to construct RNA libraries using the TruSeq RNA access library or a stranded mRNA LT sample prep kit (Illumina Inc., San Diego, CA, USA) according to the manufacturer's instructions. Quality was analyzed using an Agilent 2100 Bioanalyzer (Agilent Technologies) with an Agilent DNA 1000 Kit (Agilent Technologies). All samples were sequenced on an Illumina HiSeq2500 (Illumina), yielding an average of 38 million paired-end 100 nucleotide reads.

## Bulk RNA sequencing analysis

Reads were mapped to human reference sequence (GRCh38) by STAR aligner and read counts and transcripts per million (TPM) were quantified by RSEM. Since we observed batch bias from two library kits, we utilized the ComBat-seq algorithm to correct the batch effects[38]. To analyze the one-carbon metabolic gene expression in thyroid cancer, we used the ComplexHeatmap R package with log10 transcripts per million (TPM). We calculated the TDS using the GSVA package with the TDS gene list[8], and then analyzed the correlation between the score and one-carbon metabolic genes using the corrplot R package. Differential expression analysis (DEA) was performed using the DESeq2 R package. The results obtained via DEA were used for GSEA with the fgsea R package.

## Analysis of The Cancer Genome Atlas (TCGA) database

Genomic data for papillary thyroid carcinoma (THCA), collected as part of the TCGA project, were obtained from the TCGA Data Portal (https://tcga-data.nci.nih.gov). Gene expression data generated via RNA sequencing and clinical parameters in patients with normal ($n = 57$) and papillary thyroid carcinoma ($n = 500$), as well as clinical

data, including extrathyroidal extension and TNM stage, were analyzed.

## Sample preparation for single-cell RNA sequencing

Thyroid tissues were minced for 5 min in Hank's Balanced Salt Solution (HBSS) and digested with collagenase type 4 (400 unit/mL) (CLS-4, Worthington, Lakewood, NJ, USA) dissolved in RPMI1640 media at 37 °C for 60 min. The cells were centrifuged at 3,100 ×g for 5 min at 4 °C, and the pellets were resuspended in RPMI medium containing 10% fetal bovine serum (FBS). After mechanical dissociation by pipetting, the cell suspension was filtered through a 70-μM cell strainer and centrifuged at 3,100 × g for 5 min at 4 °C. The pellets were incubated in red blood cell lysis buffer (11814389001; Roche Diagnostics, Basel, Switzerland) for 10 min on ice. After centrifugation, the cell pellets were resuspended in FBS-free RPMI medium and filtered through a 40-μM cell strainer[64].

## Library preparation and sequencing for single-cell RNA sequencing

The concentration of the cell suspension was measured using an automated cell counter and adjusted to capture 10,000 cells/sample. Following the manufacturer's instructions, cells from the five thyroid samples were loaded according to the standard protocol of the Chromium Single Cell 3′ Reagent Kit (v3 chemistry, 10× Genomics, Pleasanton, CA, USA) or the Chromium Single Cell 5′ Reagent Kit (v1.1 chemistry). Samples were sequenced using a HiSeq X (Illumina, San Diego, CA, USA). The scRNA sequencing data were mapped, and the number of molecules per barcode was quantified using the 10× Genomics Cellranger (version 3.0.2) and the GRCh38 reference genome supplied by 10× Genomics (Supplementary Data 14). In addition, we utilized previously generated single-cell RNA sequencing data of normal thyroid tissues (GSE182416).

## Analysis of single-cell RNA sequencing data

The reads obtained from sequencing were mapped to the human reference genome (GRCh38) using the Cellranger software. Initially, each sample was processed individually using the Seurat R package[65]. Cells with a high proportion of mitochondrial reads (> 20%) or an insufficient number of detected genes (< 200) were excluded. Normalization and scaling were performed using the top 2000 variable genes. After principal component analysis (PCA) was conducted, the top 30 principal components were used for UMAP projection and Louvain clustering, following the default settings in Seurat. The cell types were assigned based on known marker genes. DoubletFinder was used to identify and annotate potential multiplets[66].

Subsequently, individual datasets were merged using the anchor-based canonical correlation analysis (CCA) integration function in Seurat[65]. This process involves the use of cross-sample pairs of cells with matched biological functions as anchors to correct batch effects. Scaling, PCA, clustering, and UMAP projections were repeated using the merged dataset. To narrow our focus to differences in tumor tissues, we excluded cells from adjacent normal tissues that were not thyroid cells. Clusters exhibiting poor quality with a relatively high mitochondrial proportion and low RNA content or containing a significant number of multiplet-marked cells, as well as clusters showing enrichment of specific cell cycles, were systematically removed. Clustering and UMAP projection steps were repeated until all low-quality clusters were eliminated.

We have noticed that when using anchor-based integration, there may be an issue with the overcorrection of the batch in thyroid-origin cells. To address this problem, we utilized the ComBat-seq algorithm to apply batch-effect correction to the entire gene expression matrix of thyroid-origin cells[38]. This ensures that comparisons across all expressed genes are fair and unbiased. We used the matrix to compare gene and pathway expression within thyroid cells.

When analyzing DEGs and pathways for each cell type, stricter criteria were applied to filter out low-quality cells (i.e., total read count <10,000 or proportion of mitochondrial RNA > 10%) to ensure robust results. After removing these cells, up to 500 cells of each cell type were randomly selected, proportionally representing the samples without introducing a significant bias. To account for batch effects in all expressed genes, including those with low expression levels, we applied the Combat-seq algorithm to the count matrix[38]. Mitochondrial genes and genes closely associated with the expression of ribosomal protein genes were excluded prior to the DEG and pathway analyses. R packages DESeq2 and fgsea were employed for DEG analysis and GSEA, using the batch-corrected count matrix[67,68]. The ranking of genes for the GSEA was defined as

$$\sqrt{\log(foldchange)^2 + (negative\ log(adjusted\ pvalue))^2} \times Direction\ of\ fold\ change\ (+1\ or\ -1),$$

using the values obtained from the DEG analysis.

To compare gene expression levels among tumor cells, log10 count per million (log10 CPM) values were extracted from the batch-corrected count matrix. TDS was calculated as the scaled mean of genes related to thyroid differentiation, as defined in a previous study[8]. For trajectory inference, the RNA velocity algorithm was utilized, with spliced and unspliced reads counted in each sample and merged into a loom file using Python scripts from velocyto[39]. Then, RNA velocity was calculated using the R script of velocyto and applied to the UMAP embedding. When identifying DEGs between TDS-high SM-low cells and TDS-low SM-high cells, the FindMarkers function in the Seurat R package was employed.

## Cell culture

BCPAP (ACC273) and 8505 C (ACC219) cells were purchased from DSMZ (Germany) in 2021. Nthy-ori 3-1, TPC-1, FRO and luciferase-expressing FRO (FRO-Luc) cells were provided by Dr. Young Joo Park and Dr. Sun Wook Cho (Seoul National University College of Medicine, Seoul, Republic of Korea). Nthy-ori 3-1, TPC-1, and FRO were authenticated by the short tandem repeat (STR) typing method using AmplFLSTR identifiler PCR Amplification kit (cat.4322288, Applied Biosystems, Foster, CA, USA), 3530xL DNA Analyzer (Applied Biosystems), and GeneMapper v5 (Applied Biosystems) on May, 2020. The results of STR profiling were summarized in Supplementary Data 15. TPC-1 cells were cultured in DMEM (Welgene, Daegu, Korea), and the other cell lines were maintained in RPMI1640 (Welgene) supplemented with 10% FBS (Gibco, CA, USA), penicillin (100 U/mL), and streptomycin (100 g/mL) (Invitrogen, MA, USA) in a humidified incubator with a 5% $CO_2$ atmosphere.

## Isotopic labeling and GC-MS analysis of $^{13}C$ metabolite enrichment from C13 metabolites

For isotopic labeling experiments, thyroid cells, including BCPAP, 8505 C, and FRO cells, were cultured in 60 mm dishes overnight. Labeling experiments were performed in the aforementioned media without serine or with serine supplementation (300 μM). The cells were then incubated for 6 h for tracing following cell harvesting. GC-MS was employed to analyze $^{13}C$ enrichment from serine into metabolites, as previously described[69,70]. Briefly, cells were washed twice with 2 mL of ice-cold isotonic saline solution (9 g/L NaCl), and metabolite extraction was performed by the addition of 250 μL of 80% ice-cold methanol solution. The cells were harvested and collected in pre-chilled tubes, followed by a medium intensity sonication with a 10 s on/off cycle for 10 min at 4 °C using a BioRuptor PICO (Diagenode). The metabolite suspensions were pelleted for 10 min at 16,400 rpm at 4 °C, and supernatants were transferred into a glass vial (Agilent Technologies) and then dried in a vacuum centrifuge (Marda). The fully dried extracts were dissolved in 10 mg/mL methoxyamine dissolved in pyridine (Sigma Aldrich), then incubated for 30 min at 37 °C, followed by derivatization by sialylation with 70 μL of N-

methyl-N-tert-butyl-dimethylsilyltrifluoroacetamide (MTBSTFA, Sigma) for 1 h at 70 °C. Samples were analyzed on an Agilent 5977 B mass selective detector coupled to an 8890 gas chromatograph (Agilent Technologies) with a 7693 A autosampler and a DB-5MS + DG capillary column (30 m plus 10 m Duraguard®, Agilent Technologies). All data were analyzed by electron ionization at 70 eV. Altogether, 1 μL of the derivatized sample was injected in splitless mode at 280 °C (inlet temperature) using helium as a carrier gas at a flow rate of 1.55 mL/min. The quadrupole was set to 150 °C using GC/MS interface at 285 °C. The oven program for all the analyses was initiated at 60 °C held for 1 min, followed by an increase at a rate of 10 °C/min to 320 °C. Data acquisition was performed in the full-scan mode (1–600 m/z). All metabolites were analyzed using previously validated authentic standards to confirm the mass spectra and retention times[70]. MassHunter Quantitative Analysis software (Agilent Technologies) was used for peak area integration. $^{13}C$ enrichment from serine into metabolites was analyzed using a previously developed algorithm, and mass isotopologue distributions (MIDs) were calculated to analyze the C13 fractional abundance of the metabolites, as previously described[71–73].

## Generation of stable cell lines

To generate stably transfected cell lines for *SHMT2* knock-down, *SHMT2* short hairpin RNA (shRNA) lentiviral particles were purchased from OriGene (USA). The 8505 C and FRO cells ($5 × 10^4$ cells/well) were seeded in 12-well plates. At 70–80% confluency, the cells were transfected with *SHMT2* shRNA lentiviral particles and polybrene (8 μg/mL) (Santa Cruz Biotechnology). After 24 h, the medium was changed and puromycin (2 μg/mL) (Sigma-Aldrich, St. Louis, MO, USA) was added for selection. At 90–100% confluency, the cells were transferred to 6-well plates. After reaching >90% confluency, the cells were transferred to a cell culture flask, and the efficiency of transfection was confirmed by western blotting.

## Cell transfection with small interfering RNA (siRNA)

To determine the effects of *SHMT2* knockdown, a Human *SHMT2* siRNA (Set of 3, HSS109732, HSS109733, HSS109734) was purchased from Thermo Fisher Scientific. BCPAP, 8505 C, and FRO cells were cultured in plates and transfected with human *SHMT2* siRNA or negative scramble siRNA (20 nM/mL) using Lipofectamine RNAiMAX transfection reagent (Invitrogen). The cells were incubated in 5% $CO_2$ at 37 °C.

## sgRNA-CRISPR/Cas9 system design and transfection

To generate a single-guide RNA (sgRNA) construct targeting *SHMT2* and a Cas9 expression construct, we designed sgRNA containing the following target sequences: SHMT2-A, CATCATCATCAT; SHMT2-B, GACGTCAGACATGTAGCCGTGGG (Bioneer, Daejeon, Korea). To establish *SHMT2* knockout cells, cells were transfected with the CRISPR plasmids (sgRNA and Cas9) using AccuFect™ transfection reagent (Bioneer). After transfection, SHMT2 knockout was confirmed in expanded colonies from single-cell clones using western blotting.

## Western blot

Cells or tissues were lysed in radioimmunoprecipitation assay (RIPA) buffer (30 mM Tris [pH 7.5], 150 mM sodium chloride, 1 mM sodium phenylmethylsulphonyl fluoride, 1 mM sodium orthovanadate, 1% Nonidet P-40, 10% glycerol, and phosphatase and protease inhibitors) containing a 1x protease inhibitor cocktail. Western blotting was performed with protein (30–50 μg) using commercially available antibodies: anti-SHMT2 (#12762, 1:1000, Cell signaling Technology, Beverley, MA, USA), anti-MTHFD2 (ab151447, 1:1000, Abcam, Cambridge, USA), anti-total OXPHOS cocktail (ab110413, 1:1000, Abcam), anti-β-actin (ab8227, 1:1000, abcam), and anti-α-tubulin (T5168,

1:1000, Sigma-Aldrich). The appropriate secondary antibodies were obtained from Bio-Rad (Hercules, California, USA). Images were acquired using the Odyssey imaging system and quantified using Image Studio Digits (LI-COR Biosciences, Lincoln, NE, USA).

## Cell viability assay

To cell viability assay in stable cell lines, cells ($1 × 10^4$ cells/well) were seeded in 96-well plates and then measured in a time-dependent manner using the water-soluble tetrazolium salt-1 (WST-1) assay (Roche Diagnostics Corporation, Indianapolis, IN, USA).

To observe the cell viability by SHIN2, cells ($1 × 10^4$ cells/well) were seeded in 96-well plates. After 24 h, the cells were treated with SHIN2 (Aobious, USA) or DMSO for 48 h. Cell viability was measured using the cell proliferation reagent WST-1 (Roche Diagnostics Corporation, Indianapolis, IN, USA). The formazan product was measured quantitatively at 450 nm using an enzyme-linked immunosorbent assay reader.

## Cell migration assay

Cell migration assays were performed in a Transwell chamber (Corning Coster, Cambridge, MA, USA) with 6.5-mm diameter polycarbonate filters (8-μm pore size). Cells ($1 × 10^5$) in serum-free medium were seeded in the upper chamber and a medium containing 10% FBS was added to the lower chamber. After 24 h, the cells were then fixed and stained with crystal violet. The non-migrating cells on the upper surface of the filter were removed using a cotton swab. Chemotaxis was quantified by counting the number of cells that migrated to the lower surface of the filter using an optical microscope (X200).

## Oxygen consumption rate

To measure the oxygen consumption rate (OCR), $2 × 10^4$ cells were seeded in each well of XF-24 plates and placed in a 5% $CO_2$ incubator at 37 °C for 24 h. The sensor cartridge was placed in XF24 calibrant solution (Agilent, Santa Clara, CA, USA) in a non-$CO_2$ incubator for 4 h before the experiment. Before measurements were made, the medium was changed to glucose-free DMEM lacking sodium bicarbonate. OCR analysis was performed at 37 °C using the XF24 analyzer (Agilent). After measurement of the basal OCR, oligomycin A (ATPase inhibitor, final concentration 2 μg/mL), CCCP (uncoupler, final concentration 5 μM), and rotenone (mitochondrial complex I inhibitor, final concentration 2 μM) were injected into each well.

## Flow cytometry analysis

The cells were seeded ($2 × 10^5$ cells/well) in a six-well plate. Then, the cells were treated with SHIN2 (20 μM) and formate (1 mM) for 24 h. Following the manufacturer's instructions for the dead cell apoptosis kit (Invitrogen), 1x annexin-binding buffer was prepared and the propidium iodide (PI) staining solution was diluted to 100 μg/mL using 1x annexin-binding buffer. The cells were resuspended in 1x annexin-binding buffer (100 μL), Alexa Fluor 488-annexin V (5 μL), and PI (1 μL, 100 μg/mL), and incubated for 15 min at room temperature. After the incubation, 1x annexin-binding buffer (400 μL) was added and gently mixed before flow cytometry analysis. Finally, flow cytometry was performed using a BD LSRFortessa flow cytometer (BD Biosciences, San Jose, CA, USA), and the data were analyzed using FlowJo software (FlowJo, LLC, Ashland, OR, USA).

## Animal experiments

All mice fed standard chow (Teklad 2018) and were housed in a specific pathogen-free animal facility (Chungnam National University Hospital Preclinical Research Center) in a controlled environment (12 h light/12 h dark cycle; humidity, 50–60%; ambient temperature, 23 °C). All animal procedures were performed in accordance with the guidelines of Institutional Animal Care at Chungnam National University (CNUH-021-A0032). Tumor growth was monitored by digital calipers, and

tumor volumes were recorded using the following formula: Volume = (longer diameter × shorter diameter$^2$)/2. If animals appeared moribund or the diameter of the tumors reached 15 mm, the mice were sacrificed. In some cases, the maximal tumor burden permitted has been exceeded the last day of measurement and the mice were immediately euthanized. All mice were euthanized with CO2 and then tissues were obtained.

To verify the effects of *SHMT2* knockdown, shSHMT2 FRO-luc or shControl FRO-luc cells ($2 \times 10^6$ cells; 1:1 ratio with Matrigel) were injected into 6-week-old male nude mice ($n = 12$ per group). After 2 weeks, tumor growth was observed every other day for two weeks using a bioluminescence imaging system (IVIS) consisting of a Lumina XRMS instrument (PerkinElmer, MA, USA), and tumor volumes were calculated using the following equation: length × width$^2$ × 0.5.)

To determine whether the SHMT inhibitor SHIN2 affected tumor growth, FRO-luc cells were diluted ($1 \times 10^6$ cells, 1:1 ratio with Matrigel) and subcutaneously injected into 6-week-old male nude mice (total volume of 100 μL) ($n = 12$ per group). After one week, SHIN2 (200 mg/kg) or vehicle ($n = 12$ per group) was administered intraperitoneally twice daily for two weeks. Tumor growth was observed using a bioluminescence imaging system. For in vivo live imaging, mice were intraperitoneally administered d-luciferin (150 mg/kg, Promega, Madison, WI, USA) and anesthetized using 2% isoflurane in 100% O$_2$ before bioluminescence imaging. The ROIs were calculated using Living Imaging software (Caliper Life Sciences, MA, USA), which drew around the tumor and measured the total radiant efficiency (photons/sec/cm$^2$/steradian [sr]).

### Histological analysis

To obtain paraffin-embedded tissue blocks, the tissue samples were fixed in 10% formalin for 16 h at room temperature. Tissues were dehydrated, incubated in xylene, and embedded in paraffin. Paraffin-embedded tissue sections (4-μm-thick) were incubated at 56 °C for 3 h. Tissue sections were deparaffinized, hydrated, washed, and stained with H&E. For immunohistochemistry (IHC), tissue sections were deparaffinized, rehydrated, and washed, antigen retrieval was performed (10 mM citrate buffer, pH 6.0), and tissue was incubated with an anti-SHMT2 antibody (PA5-32228, 1:250, Invitrogen) for 30 min at room temperature. Detection was performed using the UltraVision LP large-volume detection system (Thermo Fisher Scientific, MA, USA).

### Statistical analysis

No statistical method was used to predetermine the sample size. No data were excluded from the analyses. The experiments were not randomized. The Investigators were not blinded to allocation during experiments and outcome assessment. The statistical analyses were performed using t-tests in the R program, GraphPad Prism 9 (GraphPad Software Inc., La Jolla, CA, USA), and SPSS version 26 (IBM Corp., Armonk, NY, USA). The correlation between genes in the SGP with tumor size and TDS was calculated using Pearson's correlation. Correlation coefficients were calculated for primary tumors. Between-group differences were compared using the chi-square test, and categorical variables were expressed as percentages. Data were expressed as the mean ± SD. A two-tailed $p$-value < 0.05 was considered significant.

### Reporting summary

Further information on research design is available in the Nature Portfolio Reporting Summary linked to this article.

## Data availability

The bulk RNA sequencing data generated in this study have been deposited in the GEO database under accession number GSE213647. The information on bulk RNA sequencing data used in this study is provided in the Supplementary Data 13. The single-cell RNA sequencing data of PTC and ATC generated in this study have been deposited in the GEO database under accession number GSE232237. The single-cell RNA sequencing data of normal thyroid tissues used in this study are available in GEO database under accession code GSE182416. The information of single-cell RNA sequencing data used in this study are provided in Supplementary Data 14. The metabolomics data are deposited in supplementary data 1. Source data are provided with this paper.

## Code availability

The code for bulk RNA sequencing analysis are deposited on https://github.com/selee316/thyroid_cancer_one_carbon [https://doi.org/10.5281/zenodo.10397841][74]. In-house scripts for single-cell RNA sequencing analyses are available on GitHub (https://github.com/syparkmd/thyroid_cancer_one_carbon) and Zenodo (https://doi.org/10.5281/zenodo.10392309)[75].

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

## Acknowledgements

The biospecimens and data used in this study were provided by the Biobank of Chungnam National University Hospital, a member of the Korea Biobank Network. This research was supported by the Ministry of Trade, Industry, and Energy (MOTIE) Systemic Industrial Infrastructure and R&D Support Program (grant number P0009796), supervised by the Korea Institute for Advancement of Technology (KIAT) and the National Research Foundation of Korea (NRF) (grant numbers 2021R1C1C1011183 to Yea Eun Kang, 2022R1I1A1A01071876 to Seong Eun Lee, and 2019R1A2C1084125 to Bon Seok Koo). This work was also supported by the Korea Health Technology R&D Project through the Korea Health Industry Development Institute (KHIDI), funded by the Ministry of Health and Welfare (grant numbers HR20C0025 and HR22C1734). This work also supported by the Korean Thyroid Association Young Investigator Award 2020.

## Author contributions

Conceptualization: Y.E.K., S.K.K., and B.S.K. Conceived and designed the experiments: S.E.L., S.P., Y.E.K., and B.S.K. Performed the experiments: S.E.L., S.P., S.Y., M.A.L., S.J.Y., M.J.L., M.K.Y., and D.W.C. Analyzed the data: S.E.L., S.P., J.-Y. K., Y.E.K., B.S.K. Contributed reagents/materials/analysis tools: J.W.C., H.R.W., J.R.K., H.M.K., E.J.C., Y.J.P., Y.S., LL, SWC, S.Y.K., S.K.K., and B.S.K. Writing – original draft: S.E.L., S.P., and Y.E.K. Writing – review & editing: S.P., Y.E.K., and B.S.K.

## Competing interests

The authors declare no competing interests.

## Additional information

[1]Research Center for Endocrine and Metabolic Disease, College of Medicine, Chungnam National University, Daejeon, Republic of Korea. [2]Research Institute for Medical Sciences, College of Medicine, Chungnam National University, Daejeon, Republic of Korea. [3]GENOME INSIGHT TECHNOLOGY Inc, Daejeon, Republic of Korea. [4]Department of Otolaryngology-Head and Neck Surgery, College of Medicine, Chungnam National University, Daejeon, Republic of Korea. [5]Department of Surgery, College of Medicine, Chungnam National University, Daejeon, Republic of Korea. [6]Department of Otolaryngology-Head and Neck Surgery, Seoul National University College of Medicine, Seoul, Republic of Korea. [7]Department of Internal Medicine, Seoul National University College of Medicine, Seoul, Republic of Korea. [8]Department of Surgery, Seoul National University Bundang Hospital, Seongnam-si, Republic of Korea. [9]Department of Pathology, College of Medicine, Chungnam National University, Daejeon, Republic of Korea. [10]Department of Biochemistry, College of Medicine, Chungnam National University, Daejeon, Republic of Korea. [11]College of Pharmacy, Seoul National University, Seoul, Republic of Korea. [12]Korea Research Institute of Bioscience and Biotechnology, Deajeon, Republic of Korea. [13]Department of Bioscience, University of Science and Technology (UST), Deajeon, Republic of Korea. [14]Korea Bioinformation Center (KOBIC), Korea Research Institute of Bioscience and Biotechnology, Daejeon, Republic of Korea. [15]Department of Nutrition, School of Public Health and Management, Wenzhou Medical University, Wenzhou 325035, China. [16]Department of Biochemistry,

Chungnam National University, Daejeon, Republic of Korea. [17]Department of Biotechnology, College of Life Sciences and Biotechnology, Korea University, Seoul, Republic of Korea. [18]Department of Chemical and Biomolecular Engineering, University of California, Los Angeles, Los Angeles, USA. [19]Graduate School of Medical Science and Engineering, Korea Advanced Institute of Science and Technology, Daejeon, Republic of Korea. [20]Division of Endocrinology and Metabolism, Department of Internal Medicine, College of Medicine, Chungnam National University, Daejeon, Republic of Korea. [21]Present address: Stanford Cancer Institute, Stanford University School of Medicine, Stanford, CA, USA. [22]These authors contributed equally: Seong Eun Lee, Seongyeol Park. ✉e-mail: seonkyu@kribb.re.kr; bskoo515@cnuh.co.kr; yeeuni2200@gmail.com

