## [Peer Review File · Nature Communications]

REVIEWERS' COMMENTS:

Reviewer #1 (Remarks to the Author); expert in metabolism:

The present manuscript applies Omic technologies to examine metabolism and gene expression in a large cohort of thyroid cancer patients compared to normal controls. The scale of the effort is impressive and the use of both gene expression and metabolomics in the same samples is promising. Also positive is the use of single cell sequencing in a few samples, which cover differentiated and less differentiated thyroid cancer. Collectively the data support that SHMT2/MTHFD2 (enzymes catalyzing consecutive steps in mitochondrial serine catabolism to yield usable one-carbon units) are amplified especially in the less differentiated cancer.

On the negative side, the presentation of single cell sequencing does not provide an effective global image of the cell types in the tumors, or reveal meaningful insights beyond confirming the above.

Most problematically, claims regarding the importance of the one-carbon pathway in undifferentiated thyroid cancer are established using only a single cell line per cancer subtype and in vivo workup is very insufficient, with no tumor growth curves, no survival curves, etc.

Generally, writing is less good than typical for a Nature Communications paper. For example, in the abstract, we are told that some cells have a unique profile but this is not defined. There is hyping language like "dramatic changes" "novel" (for a much-discussed target already) and scientifically inaccurate inference of causation "decreases tumor size by up-regulating TDS markers" (No evidence is presented that the pathway suppressing tumor size runs through re-differentiation.) Some relevant references are also missing.

My general feeling is that the authors have collected an important set of information on clinical tumors without analyzing and writing with sufficient care to make the most of it, and with too limited in vivo animal follow-up to prove the functional importance claims.

Reviewer #2 (Remarks to the Author); expert in thyroid cancer:

The authors explore the potential role of serine/glycine metabolism in thyroid cancer using a combination of approaches. The authors perform metabolomics profiling of 17 normal and 17 tumor tissue by LC-MS, bulk RNA sequencing of 263 normal thyroid and 369 tumors (348 PTC, 5 PDTC, 16 ATC), and single cell RNA sequencing of cells derived from 3 normal thyroids, 7 PTCs and 5 ATC cases. The authors then specifically focus on the role of serine/glycine metabolism in thyroid tumors by examining metabolite concentrations and RNA expression patterns from these large datasets.

The authors observe increased levels of serine and glycine in tumor tissue compared to normal, as well as higher expression of SHMT2 and MTHFD2 that are involved in mitochondrial 1-carbon/folate metabolism. The authors find a positive correlation between SHMT2/MTHFD2 expression and dedifferentiation by examination of the "thyroid differentiation score" (TDS) developed by the TCGA study and observe an increased SHMT2/MTHFD2 expression in PDTC/ATC samples. Single cell RNA sequencing supports these observations in that SHMT2/MTHFD2 expression negatively correlates with TDS. The authors explore the functional role of this pathway in thyroid cancer cell proliferation using siRNA and shRNA to knock down SHMT2 in thyroid cancer cell lines and observe decreased proliferation of an ATC cell line, FRO, following expression of shRNAs targeting SHMT2. The authors then use a dual SHMT1/2 inhibitor, SHIN2 and demonstrate that SHIN2 suppressed cellular proliferation, mitochondrial respiration, and tumor growth using thyroid cancer cell lines and xenografts.

The manuscript represents a very large amount of omics data obtained from clinically relevant thyroid cancer specimens. However, the authors followed a pre-conceived hypothesis to evaluate serine/glycine/1-carbon metabolism pathways, and do not present a clear unbiased/unsupervised analysis of the data. As such, it is not clear where serine/glycine/1-carbon pathways rank with respect to altered metabolic pathways in low TDS/PDTC/ATC tumors (which are the most clinically relevant). In addition, over-expression of SHMT2/MTHFD2 has been observed in many cancer

types, therefore, it remains unclear if this manuscript has uncovered a thyroid cancer-specific role for these pathways. Finally, in the functional studies presented in Figures 4 and 5 would require the use of a larger panel of cell lines, additional controls, and rescue experiments to be able to draw clear conclusions with respect to the role of this pathway in thyroid cancer cell proliferation.

Major Criticisms:

1. An unbiased/unsupervised analysis of the omics data should be shown to determine where serine/glycine/1-carbon pathways rank.
2. The functional experiments presented in Figure 4 are worrisome for off-target shRNA effects. Only one ATC cell line, FRO, exhibits high SHMT2 expression, and the degree of knockdown by western blot doesn't correlate well with the effects on proliferation. The authors should attempt to rescue knockdown of SHMT2 by expressing a shRNA-resistant cDNA, or alternatively (and likely experimentally easier) use an orthogonal method like CRISPR to independently confirm these results. It is notable that SHMT2 doesn't appear to be essential in thyroid cancer cell lines in the DepMap (including in 8505C and CAL62 cells used in this study) and other studies have suggested somewhat redundant function of SHMT1/2 for proliferation. Another possibility would be to attempt to rescue SHMT2 knockdown by adding additional glycine or formate to the media. The use of additional PTC and ATC cell lines might also strengthen the concept that regulation of the pathway is tied directly to tumor histology/differentiation state.
3. The concentrations reported for SHIN2 appear far above the IC50 of the compound for inhibition of SHMT1/2. In Figure 5a, the authors report a dose-response starting at 10uM (10nm/mL) going to 20mM (20uM/mL). Hopefully this is simple typo of the units in the figure. However, considering an IC50 of 318nM for suppression of leukemia cell proliferation reported by Garcia-Canaveras et al, Leukemia 2021, the use of 5-20uM in experiments for the figure increases the potential for off-target effects.
4. Target engagement of SHIN2 in cell lines and ideally tumors should be examined by metabolite profiling. In addition, formate has been shown to rescue SHMT1/2 inhibition and should be used in Figure 5 to demonstrate an on-target effect of the small molecule for the assays shown.
5. Are the same changes in mitochondrial respiration observed following SHIN2 treatment also observed in the siRNA/shRNA studies shown in Figure 4? It would be important to have consistency between the small molecule and genetic studies.
6. For the xenograft tumor studies in Figure 5, tumor size and final tumor weight should be reported in addition to the bioluminescence values.

Minor Criticisms:

1. The tumor subtypes (PTC/ATC/PDTC) should be specified in the metabolomics data shown in Figure 1
2. Figure 1b is unclear and the actual data presented are too small to evaluate. The Y-axis should be labeled for the reader to better understand what data is being presented.

Novelty and technical issues

3. Figure 2f – are the tumors in the study really only 2mm? This would be considered microcarcinoma, which are generally not clinically aggressive tumors.
4. The cell plates stained for viable cells needs to be enlarged and explained in the figure legend.
5. More experimental detail should be written into the Figure 4/5 legends.
6. The manuscript is generally well-written, but would benefit from additional editing for grammar.

Reviewer #3 (Remarks to the Author); expert in transcriptomics and thyroid cancer:

Lee et al. study the relevance of SHMT2 and the one-carbon pathway in differentiated and undifferentiated tumors through different cohorts and experiments.

With mass spectrometry, they see that unlike SSP-related metabolites, L-serine and L-glycine are upregulated in 17 PTC compared to 17 normal tissues.

With bulk-RNA sequencing on 348 PTCs, 5 PDTCs, and 16 ATCs and 263 adjacent normals, they see that the expression of mitochondrial one-carbon genes, SFXN3 and serine/folate transporters was increased in tumours and negatively correlated with TDS, while the expression of cytosolic one-carbon and SSP-related genes was reduced and negatively correlated with TDS. This was confirmed in PTC from TCGA. Expression of mitochondrial one-carbon and aerobic glycolysis-involved genes was increased in dedifferentiated cancers relative to differentiated cancers. C13 glucose tracing in cell lines shows that glycolytic efficiency was higher in ATC, whereas glycolytic carbon flux did not contribute to de novo serine synthesis.

High expression of SHMT2/MTHFD2 was seen more in advanced cancers and also associated with aggressive clinical features.

Single cell RNA sequencing on 7 PTC, 5 ATC and 3 normal tissues after batch-correction recapitulated the bulk signal in the tumour cells, and suggested that higher expression SHMT2/MTHFD2 and genes related to glycolysis was seen increased in dedifferentiated tumour cells compared to differentiated tumour cells compared to normal (ATC>PTC>normal) and the opposed trend for PHGDH and SHMT1. Similar signals were seen in pathway analyses. Trajectory analysis revealed distinct trajectories for TDS-high and SM-high phenotypes.

SHMT2 protein was overexpressed in PTC and further so in ATC. The authors used siRNA and shRNA lentiviral particles to knock-down SHMT2 in cell lines, showing that it decreased cell proliferation, viability, and migration in thyroid cancer cell lines with increased effect in dedifferentiated cancers, but had no effect in normal thyroid cells.

They then use SHIN2, a small-molecule inhibitor of SHMT to explore its impact in cell lines and xenograft. It decreases cell viability and oxygen consumption rate, as well as mitochondrial complex proteins. Both a knockdown and the drug decreased tumor mass in nude mice. It also significantly increased the expression of TSHR and TTF1, but not of NIS, TPO, or TG.

Altogether, this study suggests that, as seen in other cancer types, SHMT2 could be a good candidate target for treatment in advanced thyroid cancers.

The study is clearly written, although missing important technical information for understanding and reproducibility. Some of the main results are interesting and novel for thyroid cancers. The appropriate literature is cited but the results are maybe not contrasted enough against the cited existing studies e.g. on the expression patterns of SHMT1 and SHMT2 in thyroid cancers. Importantly, the bulk and single-cell RNA-seq data presented are largely underexploited, as the study only focuses on two main families of pathways (thyroid and serine/glycine-related pathways), and the absence of more thorough analyses of these large rich datasets is underwhelming.

I also believe that the single-cell data are plagued with technical artefacts, which challenges the conclusions from these. Unfortunately, these data were not made available to reviewers.

In its current state, I cannot recommend publication of this manuscript. I propose major comments to be addressed, which I hope would improve the manuscript.

Comments.

- Only the bulk RNA-seq data of the primary tumour and normal tissues is available. Please make the rest of the data available, especially the single-cell data, which would represent a significant addition to the literature in terms of number of samples. All data should also be made available upon potential publication (not only to reviewers), so the manuscript is missing a data availability

statement with the appropriate links.

- There are quite a few important pieces of information missing from the manuscript:

* Regarding the patient tissues, what was the fixation used? Are these FFPE, fresh frozen, etc.?

This is critical information that is missing.

* According to the annotation of the data released in the repository, the tumours from the bulk RNA-seq experiment are all refractory tumours. This is not mentioned at all in the text. Could the authors comment on this and discuss it in the manuscript.

* For the single cell experiments, how did the authors optimise the dissociation protocol for the thyroid tissues?

* Which marker genes and for which cell types were used to identify cells in the single-cell RNA data. Please show these data (not only thyrocyte-related cells).

* Please show the 10X Cell Ranger report for the different batches, as it seems the data might be of poor quality but it is difficult to assess without the metrics.

- The bulk RNA-seq dataset represents a large and rich resource, especially as it includes rare subtypes PDTC and ATC. However, the authors do not analyse it to its full potential in this study, only looking at the few genes of interest. What other signals do the authors find in this dataset that would distinguish normal, PTC, PDTC and ATC; are these consistent with the literature? This also means that most of the signals could be unspecific. The authors should have a more general approach, including all pathways from different pathway databases (e.g. KEGG, REACTOME) to test the specificity of the signals they observe. Indeed, given the large transcriptomic differences between the different types of tissues, most genes are expected to be either up- or down-regulated between tissue types. Thus, dysregulation signals at a small scale (a few genes) such as discussed by the authors, could be due to chance alone, and are akin to the flip of a coin. Unless the proper controls (other pathways/genes) are used, the results are unspecific.

- The analysis of the single-cell dataset occupies a whole result section, and is important to make sure that the observed bulk signals in primary tissues come from tumour cells rather than cells from the microenvironment. However, its quality seems compromised. Bulk primary tumours should have non-tumour contaminant cell types whose transcriptomes differ drastically from each other and from thyroid cell types, leading to clearly separated clusters of cells on the UMAP. Similarly, ATC should have large genomic alterations that lead to independent clusters on the UMAP. Instead, a single "blob" of cells appear on the UMAP in this study, which is indicative of a poor-quality/failed experiment or suboptimal analyses.

- The authors use batch-correction for the single cell dataset, which is known to potentially remove important biological signals. Could the authors discuss this and show the UMAPs before/after the correction, as this could be responsible for the poor discrimination of cell clusters on the UMAP.

- Please tone down the title: "Mitochondrial SHMT2 is a 'crucial' therapeutic target in dedifferentiated thyroid cancer". From the study it seems like SHMT2 would indeed be a good potential candidate target for dedifferentiated thyroid cancers.

- 213-216 this analysis is redundant with the correlation analyses; please remove or shorten/merge.

RESPONSE TO REVIEWERS' COMMENTS

We thank all three reviewers for recognizing the importance of our work, professional evaluation, and constructive feedback. The reviewers' comments were very helpful, significantly improved the quality of our data interpretation, and enhanced our manuscript's clarity. To address all the reviewers' points, we conducted more analyses and experiments as outlined below.

1) To overcome the issues of the absence of unsupervised analysis both in bulk and single-cell RNA sequencing data, we conducted additional comparative analyses that revealed the metabolic properties distinguishing subgroups of thyroid cancer in a systematic manner (regarding **R1-G1**, **R2-G**, **R3-G**). We also uploaded whole data on GEO database to provide a valuable source to relevant researchers (GSE213647 for bulk RNA sequencing, GSE232237 for single-cell RNA sequencing).

2) Regarding reviewers' comments about functional studies, additional *in vitro* and *in vivo* experiments were conducted over the past months. We included the additional cancer cell lines and conducted complementary experiments, such as CRISPR-based knockdown and SHMT2 rescue by administering formate treatment (**R1-G2**, **R2-G**). We also repeated *in vivo* experiments to strengthen our hypothesis. Importantly, all of these additional analyses and experiments consistently support our original findings, providing us with more compelling and persuasive results that reinforce our conclusions.

All our major findings in the original manuscript are supported by additional analyses. Thanks to the reviewers' constructive comments, we feel that the revised manuscript is more comprehensive and written more clearly than the original manuscript. Our study provides compelling evidence highlighting the significance of the mitochondrial one-carbon pathway in undifferentiated thyroid cancer through a diverse array of data analyses and experimental approaches. These findings offer valuable insights into tumorigenesis and the development of novel therapeutic strategies in the field of thyroid cancer. Here we describe in detail the individual points.

POINT-BY-POINT RESPONSES

Reviewer expertise:

Reviewer #1: expert in metabolism

Reviewer #2: expert in thyroid cancer

Reviewer #3: expert in transcriptomics and thyroid cancer

Reviewers' comments:

Reviewer #1 (Remarks to the Author)

R1-General-1. The present manuscript applies Omic technologies to examine metabolism and gene expression in a large cohort of thyroid cancer patients compared to normal controls. The scale of the effort is impressive and the use of both gene expression and metabolomics in the same samples is promising. Also positive is the use of single cell sequencing in a few samples, which cover differentiated and less differentiated thyroid cancer. Collectively the data support that SHMT2/MTHFD2 (enzymes catalyzing consecutive steps in mitochondrial serine catabolism to yield usable one-carbon units) are amplified especially in the less differentiated cancer. On the negative side, the presentation of single cell sequencing does not provide an effective global image of the cell types in the tumors, or reveal meaningful insights beyond confirming the above.

R1-General-1-Answer: It is an excellent summary of our manuscript. We appreciate the reviewer for the positive appraisal of our study. To overcome the negative side, we added the global image of the cell types in the tumors (**New Figure 3a** and **New Supplementary Figure 3a**) and compared the proportions and enriched pathway between subgroups in **New Supplementary Figure 3b-d** and **New Supplementary Figure 4b**. We have uploaded all the data onto the GEO database, which will serve as a valuable resource for many researchers (GSE213647 for bulk RNA sequencing, GSE232237 for single-cell RNA sequencing).

< New Figure 3>

<New Supplementary Figure 3>

<New Supplementary Figure 4>

R1-General-2. Most problematically, claims regarding the importance of the one-carbon pathway in undifferentiated thyroid cancer are established using only a single cell line per cancer subtype and in vivo workup is very insufficient, with no tumor growth curves, no survival curves, etc.

R1-General-2-Answer: We appreciate the reviewer for the comment, which made us explore our observations. As per your comments, we additionally analyzed more than two cell lines per cancer subtype to validate our results. We used TPC-1 and BCPAP cells as PTC (differentiated thyroid cancer) cell lines, and 8505C and FRO cells as ATC (undifferentiated thyroid cancer) cell lines. The summary of the added experiments is as follows.

In vitro

Original: In **Figure 4** and **Supplementary Figure 6**, we found that knock-down of SHMT2 reduced cell viability and migration in undifferentiated thyroid cancer cells using 8505C, FRO, and BCPAP.

Revised: In **New Figure 4** and **New Supplementary Figure 7**, we compared the cell viability of shSHMT2 and control cells in a time-dependent manner in BCPAP, TPC-1, 8505C, and FRO. We also found decreased mitochondrial function and downregulation of mitochondrial complex by SHMT2 knockdown in TPC-1, 8505C, and FRO. We also identified the impact of SHMT2 using CRISPR-based knockdown (**New Supplementary Figure 8-9**). In **New Figure 5** and **New Supplementary Figure 10**, we identified the inhibition of SHMT2 was also reduced cell viability on the concentration of drug dependent manner and effects of SHMT2 inhibition were rescued after formate treatments.

In vivo

Original: In **Figure 5**, we compared the tumor volume measured via bioluminescence imaging of the ROI in BALB/c nude mice injected FRO-luc cells treated with/without SHIN2 or in relation to knockdown of SHMT2 at one time.

Revised: In **New Figure 6**, Tumor volume was measured continuously for 2 weeks at 2-3 day intervals using two measurement methods, including bioluminescence imaging of the ROI and volume measurements via Diameter and height measurement in mice injected shControl/shSHMT2-FRO-luc. We also investigated the survival rate until 4 months, however, there was no lethal case, which is thought to be a limitation of this study using the nude model.

<New Figure 4>

<New Figure 5>

<New Figure 6>

R1-General-3. Generally, writing is less good than typical for a Nature Communications paper. For example, in the abstract, we are told that some cells have a unique profile but this is not defined. There is hyping language like "dramatic changes" "novel" (for a much-discussed target already) and scientifically inaccurate inference of causation "decreases tumor size by up-regulating TDS markers" (No evidence is presented that the pathway suppressing tumor size runs through re-differentiation.) Some relevant references are also missing. My general feeling is that the authors have collected an important set of information on clinical tumors without analyzing and writing with sufficient care to make the most of it, and with too limited *in vivo* animal follow-up to prove the functional importance claims.

R1-General-3-Answer: We agree with your opinion, and it is a critical comments of our manuscript. As per your comments, we changed scientifically inaccurate expressions in the manuscript including grammar. We revised our manuscript carefully and we changed many sentences in the manuscript to avoid scientifically inaccurate inference. An example of our modification is as below.

Original:

More importantly, such dramatic changes are functionally relevant, as inhibition of SHMT2 significantly compromises mitochondrial respiration and decreases tumor size by upregulating TDS markers.

Revised:

SHMT2 inhibition significantly compromised mitochondrial respiration and decreased ATC cell proliferation and tumor size *in vitro* and *in vivo*.

Reviewer #2 (Remarks to the Author);

R2-General. The authors explore the potential role of serine/glycine metabolism in thyroid cancer using a combination of approaches. The authors perform metabolomics profiling of 17 normal and 17 tumor tissue by LC-MS, bulk RNA sequencing of 263 normal thyroid and 369 tumors (348 PTC, 5 PDTC, 16 ATC), and single cell RNA sequencing of cells derived from 3 normal thyroids, 7 PTCs and 5 ATC cases. The authors then specifically focus on the role of serine/glycine metabolism in thyroid tumors by examining metabolite concentrations and RNA expression patterns from these large datasets.

The authors observe increased levels of serine and glycine in tumor tissue compared to normal, as well as higher expression of SHMT2 and MTHFD2 that are involved in mitochondrial 1-carbon/folate metabolism. The authors find a positive correlation between SHMT2/MTHFD2 expression and dedifferentiation by examination of the “thyroid differentiation score” (TDS) developed by the TCGA study and observe an increased SHMT2/MTHFD2 expression in PDTC/ATC samples. Single cell RNA sequencing supports these observations in that SHMT2/MTHFD2 expression negatively correlates with TDS. The authors explore the functional role of this pathway in thyroid cancer cell proliferation using siRNA and shRNA to knock down SHMT2 in thyroid cancer cell lines and observe decreased proliferation of an ATC cell line, FRO, following expression of shRNAs targeting SHMT2. The authors then use a dual SHMT1/2 inhibitor, SHIN2 and demonstrate that SHIN2 suppressed cellular proliferation, mitochondrial respiration, and tumor growth using thyroid cancer cell lines and xenografts.

The manuscript represents a very large amount of omics data obtained from clinically relevant thyroid cancer specimens. However, the authors followed a pre-conceived hypothesis to evaluate serine/glycine/1-carbon metabolism pathways, and do not present a clear unbiased/unsupervised analysis of the data. As such, it is not clear where serine/glycine/1-carbon pathways rank with respect to altered metabolic pathways in low TDS/PDTC/ATC tumors (which are the most clinically relevant).

In addition, over-expression of SHMT2/MTHFD2 has been observed in many cancer types, therefore, it remains unclear if this manuscript has uncovered a thyroid cancer-specific role for these pathways. Finally, in the functional studies presented in Figures 4 and 5 would require the use of a larger panel of cell lines, additional controls, and rescue experiments to be able to draw clear conclusions with respect to the role of this pathway in thyroid cancer cell proliferation.

R2-General-Answer:

We appreciate the reviewer's helpful comments and suggestions on our study. Based on the reviewer's comments, we have analyzed the data in a more systematic way. Moreover, we conducted additional experiments with more than two cell lines per cancer subtype to validate our results. We also conducted more *in vivo* experiments to support our conclusion.

Major Criticisms:

R2-Q01. An unbiased/unsupervised analysis of the omics data should be shown to determine where serine/glycine/1-carbon pathways rank.

R2-A01. We appreciate the reviewer's insightful comment. We added the results of unbiased analyses encompassing all biological pathways rather than focusing only on the serine/glycine metabolic pathway (SGP).

We first compared total metabolites between normal and tumor tissues. Among all measured 216 metabolites, we found that amino acids, including L-glycine and L-serine, and the metabolites of purine metabolism (e.g., AMP, GMP, and hypoxanthine) were significantly increased in the tumors suggesting an impact of amino acid metabolism and relevant nucleotide metabolism on thyroid cancer (**New Figure 1b** and **New Supplementary Table 1**).

Next, we performed differentially expressed genes (DEGs) and gene-set enrichment analysis (GSEA) using transcriptomics data to compare tumor vs. normal and TDS-high vs. TDS-low tumors. These analyses revealed multiple distinct metabolic pathways. Notably, glycine, serine, and threonine metabolism were significantly enriched in both comparisons (**New Figure 1d and e**). Among the genes within the metabolic pathways, many SGP-related genes, such as SHMT2, MTHFD2, PSPH, and TYMS, were highly ranked DEGs with significant adjusted p -value and fold changes in the analyses (**New Figure 1f, g and New Supplementary Figure 1g**). Collectively, these multi-omics data indicate the importance of SGP in thyroid cancer dedifferentiation.

<New Figure 1>

<New Supplementary Figure 1>

R2-Q02. The functional experiments presented in Figure 4 are worrisome for off-target shRNA effects. Only one ATC cell line, FRO, exhibits high SHMT2 expression, and the degree of knockdown by western blot doesn't correlate well with the effects on proliferation. The authors should attempt to rescue knockdown of SHMT2 by expressing a shRNA-resistant cDNA, or alternatively (and likely experimentally easier) use an orthogonal method like CRISPR to independently confirm these results. It is notable that SHMT2 doesn't appear to be essential in thyroid cancer cell lines in the DepMap (including in 8505C and CAL62 cells used in this study) and other studies have suggested somewhat redundant function of SHMT1/2 for proliferation. Another possibility would be to attempt to rescue SHMT2 knockdown by adding additional glycine or formate to the media. The use of additional PTC and ATC cell lines might also strengthen the concept that regulation of the pathway is tied directly to tumor histology/differentiation state.

R2-A02. Thank you for your valuable feedback. To investigate the involvement of SHMT2 in thyroid cancer, we conducted additional experiments using PTC cell lines (TPC-1 and BCPAP) and ATC cell

lines (8505C and FRO). To ensure the reproducibility of our findings, we employed the CRISPR/Cas9 system, along with shRNA and siRNA in all cell lines. We observed that decreasing SHMT2 resulted in reduced cell viability and increased apoptosis. However, we observed that these effects were reversed with the use of formate (**New Figure 4** and **New Supplementary Figure 6-9**). Furthermore, SHIN2, an inhibitor of SHMT1/2, also resulted in reduced cell viability and apoptosis, but this was rescued by the use of formate (**New Figure 5** and **New Supplementary Figure 10**).

<New Figure 4>

<New Figure 5>

R2-Q03. The concentrations reported for SHIN2 appear far above the IC₅₀ of the compound for inhibition of SHMT1/2. In Figure 5a, the authors report a dose-response starting at 10uM (10nm/mL) going to 20mM (20uM/mL). Hopefully this is simple typo of the units in the figure. However, considering an IC₅₀ of 318nM for suppression of leukemia cell proliferation reported by Garcia-Canaveras et al, Leukemia 2021, the use of 5-20uM in experiments for the figure increases the potential for off-target effects.

R2-A03. Thank you for your detailed comment. We repeated the experiments using a dose of 10nM as a starting point and found that the efficacy of SHIN2 depends on the amount administered (**New Supplementary Figure 10b-d**). Our results show that cell viability was considerably reduced upon exposure to 10nM of SHIN2.

<New Supplementary Figure 10>

R2-Q04. Target engagement of SHIN2 in cell lines and ideally tumors should be examined by metabolite profiling. In addition, formate has been shown to rescue SHMT1/2 inhibition and should be used in Figure 5 to demonstrate an on-target effect of the small molecule for the assays shown.

R2-A04. We appreciate your constructive comment. To evaluate SHIN2 target engagement in undifferentiated thyroid cancer, we followed a previous assay based on the continuous infusion of tracer amounts of [U-¹³C₃]-serine (which contains three ¹³C atoms and is accordingly M+3) and measured serine and glycine labeling by mass spectrometry (**New Figure 5a** and **New Supplementary Figure 10a**). SHIN2 treatment reduced M+2 glycine, M+1 serine, and M+2 serine levels in all thyroid cancer cell lines. We also added the formate rescue experiments and identified the SHMT2 inhibition was rescued by formate (**New Figure 5b-f** and **New Supplementary Figure 11**).

<New Figure 5>

R2-Q05. Are the same changes in mitochondrial respiration observed following SHIN2 treatment also observed in the siRNA/shRNA studies shown in Figure 4? It would be important to have consistency between the small molecule and genetic studies.

R2-A05. We appreciate your feedback regarding consistency. We conducted a measurement of the oxygen consumption rate (OCR) on thyroid cell lines that had undergone SHMT2 reduction by shRNA. The results showed a decrease in mitochondrial respiration in these cell lines (**New Figure 4c, f** and **New Supplementary Figure 7h**).

<New Figure 4>

R2-Q06. For the xenograft tumor studies in Figure 5, tumor size and final tumor weight should be reported in addition to the bioluminescence values.

R2-A06. Thank you for your helpful suggestion. We observed tumor weight and volume every two days using bioluminescence imaging and calculated tumor volume (length × width² × 0.5). Tumor mass and volume were gradually reduced in shSHMT2-FRO-luc cells-injected mice (**New Figure 6**).

<New Figure 6>

Minor Criticisms:

R2-Q07. The tumor subtypes (PTC/ATC/PDTC) should be specified in the metabolomics data shown in Figure 1

R2-A07. The metabolomics data presented in **New Figure 1b** was obtained from normal (n=17) and paired tumor (n=17) tissues from only patients with PTC. While it would have been ideal to perform metabolomics analysis on different tumor subtypes, obtaining tissue samples from refractory cancers like ATC was challenging. We added these in manuscript as below.

“Among the measured 216 metabolites, 51 metabolites were significantly elevated while 10 metabolites were significantly decreased in tumor samples from **PTC patients** compared to adjacent normal tissues (Supplementary Tables 1 and 2, Fig. 1b).”

R2-Q08. Figure 1b is unclear and the actual data presented are too small to evaluate. The Y-axis should be labeled for the reader to better understand what data is being presented. Novelty and technical issues

R2-A08. Thank you for your comment. We appreciate your feedback and agree that the original figure may have been difficult to understand. As a result, we have replaced **Original Figure 1b** with a volcano plot that compares the abundance of metabolites in tumor and normal tissues (**New Figure 1b**). This new plot has revealed the significance of amino acid and nucleotide metabolism in thyroid cancer, among all the metabolites that were analyzed.

<Original Figure 1b>

<New Figure 1b>

R2-Q09. Figure 2f – are the tumors in the study really only 2mm? This would be considered microcarcinoma, which are generally not clinically aggressive tumors.

R2-A09. Thank you for bringing to our attention the typo in our figure. We have made the necessary correction by changing the Y-axis unit from “mm” to “cm”.

R2-Q10. The cell plates stained for viable cells needs to be enlarged and explained in the figure legend.

R2-A10. Thank you for your advice. We have made the enlarged picture and added the scale bar in the new pictures.

<New Supplementary Figure 7>

<New Supplementary Figure 8>

<New Supplementary Figure 9>

R2-Q11. More experimental detail should be written into the Figure 4/5 legends.

R2-A11. Thank you for your comment. We have provided more detailed methods in the figure legends and Methods section.

R2-Q12. The manuscript is generally well-written, but would benefit from additional editing for grammar.

R2-A12. Thank you for your helpful suggestion. We have revised the grammar throughout the entire text.

Reviewer #3 (Remarks to the Author);

R3-General. Lee et al. study the relevance of SHMT2 and the one-carbon pathway in differentiated and undifferentiated tumors through different cohorts and experiments. With mass spectrometry, they see that unlike SSP-related metabolites, L-serine and L-glycine are upregulated in 17 PTC compared to 17 normal tissues. With bulk-RNA sequencing on 348 PTCs, 5 PDTs, and 16 ATCs and 263 adjacent normals, they see that the expression of mitochondrial one-carbon genes, SFXN3 and serine/folate transporters was increased in tumours and negatively correlated with TDS, while the expression of cytosolic one-carbon and SSP-related genes was reduced and negatively correlated with TDS. This was confirmed in PTC from TCGA. Expression of mitochondrial one-carbon and aerobic glycolysis-involved genes was increased in dedifferentiated cancers relative to differentiated cancers. C13 glucose tracing in cell lines shows that glycolytic efficiency was higher in ATC, whereas glycolytic carbon flux did not contribute to de novo serine synthesis. High expression of SHMT2/MTHFD2 was seen more in advanced cancers and also associated with aggressive clinical features.

Single cell RNA sequencing on 7 PTC, 5 ATC and 3 normal tissues after batch-correction recapitulated the bulk signal in the tumour cells, and suggested that higher expression SHMT2/MTHFD2 and genes related to glycolysis was seen increased in dedifferentiated tumour cells compared to differentiated tumour cells compared to normal (ATC>PTC>normal) and the opposed trend for PHGDH and SHMT1. Similar signals were seen in pathway analyses. Trajectory analysis revealed distinct trajectories for TDS-high and SM-high phenotypes.

SHMT2 protein was overexpressed in PTC and further so in ATC. The authors used siRNA and shRNA lentiviral particles to knock-down SHMT2 in cell lines, showing that it decreased cell proliferation, viability, and migration in thyroid cancer cell lines with increased effect in dedifferentiated cancers, but had no effect in normal thyroid cells.

They then use SHIN2, a small-molecule inhibitor of SHMT to explore its impact in cell lines and xenograft. It decreases cell viability and oxygen consumption rate, as well as mitochondrial complex proteins. Both a knockdown and the drug decreased tumor mass in nude mice. It also significantly increased the expression of TSHR and TTF1, but not of NIS, TPO, or TG.

Altogether, this study suggests that, as seen in other cancer types, SHMT2 could be a good candidate target for treatment in advanced thyroid cancers. The study is clearly written, although missing important technical information for understanding and reproducibility. Some of the main results are interesting and novel for thyroid cancers. The appropriate literature is cited but the results are maybe not contrasted enough against the cited existing studies e.g. on the expression patterns of SHMT1 and SHMT2 in thyroid cancers.

Importantly, the bulk and single-cell RNA-seq data presented are largely underexploited, as the study only focuses on two main families of pathways (thyroid and serine/glycine-related pathways), and the absence of more thorough analyses of these large rich datasets is underwhelming.

I also believe that the single-cell data are plagued with technical artefacts, which challenges the conclusions from these. Unfortunately, these data were not made available to reviewers.

In its current state, I cannot recommend publication of this manuscript. I propose major comments to be addressed, which I hope would improve the manuscript.

R3-General-Answer:

We are grateful to the reviewer for their comment, as it prompted us to conduct additional analyses and enhance our research in multiple ways. We included the results of analyses from bulk and single-cell RNA-seq data that cover broad biological pathways rather than just focusing on the serine/glycine metabolic pathway (SGP). This information gives a better understanding of thyroid cancer and explains

our rationale to research SGP in this specific type of cancer. We conducted rigorous filtering processes of single-cell data to exclude possible bias by technical artifacts. We are also providing all the data we used so that other researchers can find additional meaning in these large datasets. We believe that our dataset will be a valuable resource for the thyroid cancer community.

Comments.

R3-Q01. Only the bulk RNA-seq data of the primary tumour and normal tissues is available. Please make the rest of the data available, especially the single-cell data, which would represent a significant addition to the literature in terms of number of samples. All data should also be made available upon potential publication (not only to reviewers), so the manuscript is missing a data availability statement with the appropriate links.

R3-A01. Thank you for comments. All raw data for this study were deposited in the GEO database. The accession number for the bulk RNA sequencing dataset is GSE213647 (secure token: avwpkueoznwdzux). The accession numbers for single-cell RNA sequencing are GSE232237 (secure token: mnijueaijgnnih).

R3-Q02. There are quite a few important pieces of information missing from the manuscript: Regarding the patient tissues, what was the fixation used? Are these FFPE, fresh frozen, etc.? This is critical information that is missing.

R3-A02. We are grateful for the reviewer's helpful comments. Out of the total 632 bulk RNA-seq samples, 552 samples are fresh frozen while the remaining 80 samples are FFPE. We have created a **New Supplementary Table 18** that provides detailed information regarding the samples. We also add the information in **New Methods**.

<New Supplementary Table 18>

Sample ID	GEO ID	Tissue type	Fixation type	Library kit	Dx_year	Age	Sex
1-N_RNA-seq	GSM6590734	Normal	Fresh Frozen	stranded mRNA LT sample prep kit		2019	32 F
1-T_RNA-seq	GSM6590997	PTC	Fresh Frozen	stranded mRNA LT sample prep kit		2019	32 F
10-N_RNA-seq	GSM6590742	Normal	Fresh Frozen	stranded mRNA LT sample prep kit		2019	60 F
10-T_RNA-seq	GSM6591004	PTC	Fresh Frozen	stranded mRNA LT sample prep kit		2019	60 F
100N_RNA-seq	GSM6590794	Normal	Fresh Frozen	TruSeq RNA access library		2007	49 F
100T_RNA-seq	GSM6591084	PTC	Fresh Frozen	TruSeq RNA access library		2007	49 F
101-T-P_RNA-seq	GSM6591085	PTC	Fresh Frozen	TruSeq RNA access library		2007	45 F
101N_RNA-seq	GSM6590795	Normal	Fresh Frozen	TruSeq RNA access library		2007	45 F
102-T_P_RNA-seq	GSM6591086	PTC	Fresh Frozen	TruSeq RNA access library		2007	31 F
103-T_P_RNA-seq	GSM6591087	PTC	Fresh Frozen	TruSeq RNA access library		2007	34 M
104-T_P_RNA-seq	GSM6591088	PTC	Fresh Frozen	TruSeq RNA access library		2007	27 F
105T_RNA-seq	GSM6591089	PTC	Fresh Frozen	TruSeq RNA access library		2007	34 F
106-T-P_RNA-seq	GSM6591090	PTC	Fresh Frozen	TruSeq RNA access library		2007	23 F
107-T-P_RNA-seq	GSM6591091	PTC	Fresh Frozen	TruSeq RNA access library		2007	20 F
108-T_P_RNA-seq	GSM6591092	PTC	Fresh Frozen	TruSeq RNA access library		2007	79 F
108N_RNA-seq	GSM6590796	Normal	Fresh Frozen	TruSeq RNA access library		2007	79 F
109T-P_RNA-seq	GSM6591093	PTC	Fresh Frozen	TruSeq RNA access library		2007	56 F
11-B_RNA-seq	GSM6591005	PTC	Fresh Frozen	stranded mRNA LT sample prep kit		2019	64 F
11-N_RNA-seq	GSM6590743	Normal	Fresh Frozen	stranded mRNA LT sample prep kit		2019	64 F
110-T_P_RNA-seq	GSM6591094	PTC	Fresh Frozen	TruSeq RNA access library		2007	49 F
110N_RNA-seq	GSM6590797	Normal	Fresh Frozen	TruSeq RNA access library		2007	49 F
111T-P_RNA-seq	GSM6591095	PTC	Fresh Frozen	TruSeq RNA access library		2007	44 F
112-T_RNA-seq	GSM6591096	PTC	Fresh Frozen	stranded mRNA LT sample prep kit		2007	55 F
112-T_P_RNA-seq	GSM6591097	PTC	Fresh Frozen	stranded mRNA LT sample prep kit		2007	43 F

R3-Q03. According to the annotation of the data released in the repository, the tumours from the bulk RNA-seq experiment are all refractory tumours. This is not mentioned at all in the text. Could the authors comment on this and discuss it in the manuscript.

R3-A03. We appreciate the reviewer's detailed comment. All the samples we analyzed in this study were treatment-naive primary tumors that were obtained by surgical resection. We added the information to **New Methods**. In addition, we also provided accurate information on the tumor type GEO metadata table and a **New Supplementary Table 18**.

R3-Q04. For the single cell experiments, how did the authors optimise the dissociation protocol for the thyroid tissues?

R3-A04. Our team has devised a successful method for dissociating human thyroid tissues in preparation for single-cell RNA sequencing. We have documented the protocol in Endocrinology and Metabolism, with the article titled "Efficient Dissociation Protocol for Generation of Single Cell Suspension from Human Thyroid Tissue for Single Cell RNA Sequencing" (PMID: 36065649). We have utilized this same protocol in our current research.

R3-Q05. Which marker genes and for which cell types were used to identify cells in the single-cell RNA data. Please show these data (not only thyrocyte-related cells).

R3-A05. We appreciate the critical feedback from the reviewer. We have made revisions to the single-cell datasets to provide a more inclusive view of the tissue landscape, rather than solely focusing on the tumor cells. Through the utilization of established marker genes such as TG, COL1A1, CDH5, and PTPRC, we have been able to categorize the various cell types within each cluster (**New Figure 3a** and **New Supplementary Figure 3a**). We also uploaded all raw data of single-cell RNA sequencing to GEO with the accession number GSE232237.

<New Supplementary Figure 3>

R3-Q06. Please show the 10X Cell Ranger report for the different batches, as it seems the data might be of poor quality but it is difficult to assess without the metrics.

R3-A06. We understand that the reviewer may have doubts about the quality of single-cell data, since we only showed tumor cell profiles. However, please refer to **New Supplementary Figure 3a**, where we demonstrate decent quality single-cell data encompassing all cell types present in thyroid cancer tissues. Multi-step filtering methods were employed to exclude low-quality cells, multipliers, and cell cycle-biased cells, ultimately yielding a dataset comprising 88,008 cells from 15 tissues (**New Methods**).

As an initial filtering process, cells with a high proportion of mitochondrial reads (>20%) or an insufficient number of detected genes (<200) were excluded in individual samples. “DoubletFinder” was also used to identify and annotate potential multipliants. Subsequently, individual datasets were merged using the integration function in “Seurat”. Clusters exhibiting poor quality with a relatively high mitochondrial proportion and low RNA content or containing a significant number of multipliant-marked cells, as well as clusters showing enrichment of specific cell cycles, were systematically removed. Clustering and UMAP projection steps were repeated until all low-quality clusters were eliminated. As per the reviewer’s request, we have uploaded all 10x Cellranger reports. You can find them at the following link (<https://www.dropbox.com/sh/54xdleb6myq8xyi/AABF5uZxlm7RjcuSf01HXGLLa?dl=0>). It’s important to note that these reports include all data before the aforementioned filtering steps are implemented.

<Figure R3GA1>

This figure is intended for the reviewer to demonstrate the well-controlled quality control status of all included cells. Percent.mt, Percent of mitochondrial genes; nFeature_RNA, Number of expressed genes.

R3-Q07. The bulk RNA-seq dataset represents a large and rich resource, especially as it includes rare subtypes PDC and ATC. However, the authors do not analyse it to its full potential in this study, only looking at the few genes of interest. What other signals do the authors find in this dataset that would distinguish normal, PTC, PDC and ATC; are these consistent with the literature? This also means that most of the signals could be unspecific. The authors should have a more general approach, including all pathways from different pathway databases (e.g. KEGG, REACTOME) to test the specificity of the signals they observe. Indeed, given the large transcriptomic differences between the different types of tissues, most genes are expected to be either up- or down-regulated between tissue types. Thus, dysregulation signals at a small

scale (a few genes) such as discussed by the authors, could be due to chance alone, and are akin to the flip of a coin. Unless the proper controls (other pathways/genes) are used, the results are unspecific.

R3-A07. We appreciate the reviewer's constructive feedback. As a response, we have included the results of general analyses that cover all biological pathways instead of solely focusing on the serine/glycine metabolic pathway (SGP). Using this approach, we could reveal the differential pathways between cancer types and gradually narrow our focus to SGP. Then, the results we made regarding SGP through bulk RNA-seq were confirmed by independent TCGA dataset, and the consistent results were evident in the single-cell RNA sequencing data. The details of the newly added results are described below.

(1) Additional analyses in metabolomics and bulk RNA-seq

We first compared total metabolites between normal and tumor tissues. Among all measured 216 metabolites, we found that amino acids, including L-glycine and L-serine, and the metabolites of purine metabolism (e.g., AMP, GMP, and hypoxanthine) were significantly increased in the tumors suggesting an impact of amino acid metabolism and relevant nucleotide metabolism on thyroid cancer (**New Figure 1b** and **New Supplementary Table 1**).

Next, we performed differentially expressed genes (DEGs) and gene-set enrichment analysis (GSEA) using transcriptomics data to compare tumor vs. normal and TDS-high vs. TDS-low tumors. These analyses revealed multiple distinct metabolic pathways. Notably, glycine, serine, and threonine metabolism were significantly enriched in both comparisons (**New Figure 1d** and **e**). Among the genes within the metabolic pathways, many SGP-related genes, such as SHMT2, MTHFD2, PSPH, and TYMS, were highly ranked DEGs with significant adjusted *p*-value and fold changes in the analyses (**New Figure 1f, g** and **New Supplementary Figure 1g**). Collectively, these multi-omics data indicate the importance of SGP in thyroid cancer dedifferentiation.

We also compared the expression of various histologic types of tumors (PTC, PDC, and ATC) as supplementary results (**New Supplementary Figure 1a-f**). We found that certain metabolic pathways for amino acids were enriched at the top differential pathways, along with the cell cycle and DNA replication.

<New Figure 1>

<New Supplementary Figure 1>

(2) Additional analyses in single-cell RNA sequencing

We have reprocessed all the single-cell datasets to display a comprehensive view of the tissue landscape, instead of just focusing on the tumor cells. For this purpose, we have re-performed the filtering process to obtain clean and high-quality cells in all cell types (**New Figure 3a** and **New Supplementary Figure 3a**). The detailed filtering process is described in **New Methods** and **R3-A06**.

We compare the tumor microenvironment between PTC and ATC groups using the single-cell RNA sequencing dataset. A high proportion of fibroblast and macrophage cells were found in ATCs, whereas PTCs were dominated by thyroid cancer cells. The finding was validated in bulk RNA-seq using CIBERSORTx (**New Supplementary Figure 3b** and **c**). In addition, we utilized GSEA to detect pathways that were enriched in PTC or ATC in each cell type. We found that genes linked to oxidative phosphorylation and antigen presentation were considerably enriched in ATC in multiple cell types. Conversely, genes associated with gap junctions were enriched in PTC (**New Supplementary Figure 3d**). The fact that these common pathways are present in various cell types emphasizes the existence of unique cellular networks for each cancer type, which requires further examination.

< New Figure 3 >

< New Supplementary Figure 3 >

R3-Q08. The analysis of the single-cell dataset occupies a whole result section, and is important to make sure that the observed bulk signals in primary tissues come from tumour cells rather than cells from the microenvironment. However, its quality seems compromised. Bulk primary tumours should have non-tumour contaminant cell types whose transcriptomes differ drastically from each other and from thyroid cell types, leading to clearly separated clusters of cells on the UMAP. Similarly, ATC should have large genomic alterations that lead to independent clusters on the UMAP. Instead, a single "blob" of cells appear on the UMAP in this study, which is indicative of a poor-quality/failed experiment or suboptimal analyses.

R3-A08. Thank the reviewer's comments. As mentioned in **R3-A06**, we reanalyzed single-cell RNA sequencing data with all cell types we detected. We conducted multiple rounds of QC check and cell filtering process, which resulted in decent quality data encompassing various cell types (**New Figure 3a** and **New Supplementary Figure 3a**). The detailed processes are described in **R3-A06** and **New Methods**. We observed improvement in the separation of tumor cell clusters after using different batch correction methods for thyroid-origin cells. Further details can be found in the next item (**R3-A09**).

R3-Q09. The authors use batch-correction for the single cell dataset, which is known to potentially remove important biological signals. Could the authors discuss this and show the UMAPs before/after the correction, as this could be responsible for the poor discrimination of cell clusters on the UMAP.

R3-A09. We really appreciate this comment. After conducting a thorough process of filtering and quality control for the cells, we utilized the CCA-based integration method from the Seurat R package. Although it worked well for immune and stromal cells, it tended to over-correct the thyroid-origin cells (**Original Figure 3a** and **New Supplementary Figure 4a**). This was observed by the lack of separation between the normal and tumor clusters. However, by implementing Combat-seq solely for the thyroid-origin cells, we were able to achieve better results in separating the normal and tumor clusters, as shown in the

New Figure 3b.

< New Figure 3 >

In order to achieve balanced cell counts across samples, a total of 100 cells were randomly chosen for each sample. These cells were then utilized for conducting PCA, with the resulting PC1 and PC2 values used for cell plotting.

< Revised >

Subsequently, we compared the expression of thyroid-origin cells among the groups to identify the intrinsic differences in tumor cells. It has come to our attention that when utilizing anchor-based integration, there is a tendency for overcorrection of the batch in thyroid-origin cells (**New Supplementary Figure 4a**). To overcome this problem, we employed the ComBat-seq algorithm to perform batch-effect correction on the entire gene expression matrix of thyroid-origin cells, ensuring unbiased comparisons across all expressed genes. This process made a better distinction between PTC, ATC, and normal thyroid cells (**New Figure 3b**).

R3-Q10. Please tone down the title: "Mitochondrial SHMT2 is a 'crucial' therapeutic target in dedifferentiated thyroid cancer". From the study it seems like SHMT2 would indeed be a good potential candidate target for dedifferentiated thyroid cancers.

R3-A10. Thank the reviewer's sincere comments. As per your feedback, we have modified the title to "Unraveling the role of the mitochondrial one-carbon pathway in undifferentiated thyroid cancer by a comprehensive multi-omics approach".

R3-Q11. 213-216 this analysis is redundant with the correlation analyses; please remove or shorten/merge.

R3-A11. Thanks. We shorten the description as below.

<Original>

Next, we divided subjects into two groups based on the TDS score to characterize the SGP expression

pattern: TDS-high (TDS>0) and TDS-low (TDS<0) (**Fig. 2d** and **Supplementary Fig. 2c**). The expression of *SHMT2* (P<0.001), *MTHFD2* (P<0.001), *SFXN3* (P<0.001), and serine/folate transporters (e.g., *SLC1A4*, P<0.001; *SLC1A5*, P<0.001; and *FOLH1*, P<0.001) was increased in the TDS-low group compared to the TDS-high group (**Fig. 2d**). In contrast, *PHGDH* (P<0.001) and *SHMT1* (P<0.001) were significantly reduced in the TDS-low group compared to the TDS-high group (**Fig. 2d**).

<Revised>

Next, we divided tumors into two groups based on the TDS scores to characterize the SGP expression pattern: TDS-high (TDS>0) and TDS-low (TDS<0) (**Fig. 2d** and **Supplementary Fig. 2c**). The differences in expression between the groups were well consistent with the results of correlation analyses.

REVIEWERS' COMMENTS

Reviewer #1 (Remarks to the Author):

This excellent revision massively improves what was initially an extensive but not focused or compelling manuscript. Now the text and figures clearly make the key points in a compelling manner. The small molecule therapeutic benefit in Fig 6 is quite encouraging. The authors should add a citation to the original Nature paper that showed impact of SHMT2 loss on respiratory complex expression.

Reviewer #2 (Remarks to the Author):

The authors have improved the manuscript by making significant changes to the data presentation and by performing new experiments. However, the premise that ATC cells are selectively sensitive to targeting SHMT2 isn't strongly supported by the data as both PTC cell lines and ATC cell lines respond similarly to targeting this pathway.

1. I appreciate the sincere effort by the authors to use orthogonal methods to test their hypotheses, including adding CRISPR/Cas9 methods. However, the effects of targeting SHMT2 using shRNAs, SHIN2, or CRISPR/Cas9, are modest overall (with the exception of the one xenograft experiment (Figure 6) which is using only a single shRNA. In addition, the effect of targeting SHMT2 in PTC or ATC cells appears similar, somewhat undermining the overall premise of the paper. For example, comparing the response of PTC cell lines (BCPAP, TPC-1) to ATC cell lines (8505C, FRO) in Figures S7-9 seems to show relatively similar impact of targeting SHMT2 in both types of cell lines. In addition, Figure S10 shows that one PTC cell line (BCPAP), and one ATC cell line (FRO) is sensitive to SHIN2, and conversely one PTC (TPC-1) and one ATC (8505C) cell lines is resistant.

2. The axes in Figure 5b-d, Figure S10 are somewhat deceptively labeled to amplify the effect of SHIN2 on cells, and should be reset to a baseline of zero instead of 60 or 70%.

3. Figure 6 shows a strong effect on xenograft growth of suppressing SHMT2 with a single shRNA, but concern for off targets in this xenograft experiment remain since only a single shRNA without a rescue experiment was used. This is of additional concern considering the relatively modest effects of targeting SHMT2 or treatment with SHIN2 in cell culture. Evidence of selectivity for ATC xenografts vs PTC xenografts might also strengthen the conclusions of this manuscript.

Minor points:

1. Please add the concentration of SHIN2 used for each experiment in all of the figures.

Reviewer #3 (Remarks to the Author):

I thank and commend the authors for the thorough responses to the comments, the replies to which are satisfying to me. The quality of the single-cell data indeed seems very good.

However, regarding my first major comment on availability: I found that downloading the data available on GEO with the provided token only gave access to the gene counts, not to the raw data.

This is a sizable and important dataset for the field. The authors should really make the raw data available, first for reproducibility as per the Nature portfolio editorial guidelines; and second and as importantly, because its availability for reuse is critical to make this study of interest to a large audience.

RESPONSE TO REVIEWERS' COMMENTS

Reviewer #1 (Remarks to the Author):

This excellent revision massively improves what was initially an extensive but not focused or compelling manuscript. Now the text and figures clearly make the key points in a compelling manner. The small molecule therapeutic benefit in Fig 6 is quite encouraging. The authors should add a citation to the original Nature paper that showed impact of SHMT2 loss on respiratory complex expression.

Answer to R1:

Thank you for your valuable comments. We added the reference in line introduction, results, and discussion as below.

Introduction

... Recently, SHMT-dependent serine catabolism was demonstrated to be the main source of one-carbon units and was critical for maintaining cellular redox control under stress conditions³⁰. Moreover, SHMT2 is critical for mitochondrial respiration and oxidative phosphorylation by stabilizing mitochondrial translation³¹.

Results.

... As serine catabolism by SHMT2 is required for efficient cellular respiration and the assembly of complex I in the respiratory chain^{31, 40}, we analyzed the effects of SHMT2 on mitochondrial respiration.

...

Discussion

... Other studies have shown that serine catabolism by SHMT2 is required to maintain mitochondrial respiration and translation by providing NADH and formylmethionyl-tRNAs^{31, 40, 62}, suggesting an association between mitochondrial serine catabolism and the modulation of the OXPHOS system. ...

Reviewer #2 (Remarks to the Author):

The authors have improved the manuscript by making significant changes to the data presentation and by performing new experiments. However, the premise that ATC cells are selectively sensitive to targeting SHMT2 isn't strongly supported by the data as both PTC cell lines and ATC cell lines respond similarly to targeting this pathway.

1. I appreciate the sincere effort by the authors to use orthogonal methods to test their hypotheses, including adding

CRISPR/Cas9 methods. However, the effects of targeting SHMT2 using shRNAs, SHIN2, or CRISPR/Cas9, are modest overall (with the exception of the one xenograft experiment (Figure 6) which is using only a single shRNA. In addition, the effect of targeting SHMT2 in PTC or ATC cells appears similar, somewhat undermining the overall premise of the paper. For example, comparing the response of PTC cell lines (BCPAP, TPC-1) to ATC cell lines (8505C, FRO) in Figures S7-9 seems to show relatively similar impact of targeting SHMT2 in both types of cell lines. In addition, Figure S10 shows that one PTC cell line (BCPAP), and one ATC cell line (FRO) is sensitive to SHIN2, and conversely one PTC (TPC-1) and one ATC (8505C) cell lines is resistant

Answer to R2 Q1:

Thank you for your detailed and valuable feedback. We acknowledge that in some experiments, the difference between PTC and ATC cell lines may not be significant. However, we believe that the general trend is consistent. Moreover, our bioinformatic results using a large number of human tissues show a contrast between PTC and ATC, which can complement our conclusion. In light of your feedback, we have revised the manuscript to tone down our description, and added a limitation in the Discussion section to address the concerns.

Change in Results

Before> These results suggest the importance of the mitochondrial folate cycle in thyroid cancer and that the inhibition of SHMT2 can more effectively regulate tumor aggressiveness in undifferentiated thyroid cancer.

After> These results suggest the importance of the mitochondrial folate cycle in thyroid cancer and that the inhibition of SHMT2 can effectively regulate tumor aggressiveness in thyroid cancer.

Add in Discussion

... It's worth noting that *SHMT2* inhibition also significantly affects PTC cell lines in some of our experiments. This may be due to variations in the cell lines used or low specificity of gene inhibition. For more consistent results with human omics data, future studies should incorporate a wider range of cell lines and provide detailed characterizations of each.

2. The axes in Figure 5b-d, Figure S10 are somewhat deceptively labeled to amplify the effect of SHIN2 on cells, and should be reset to a baseline of zero instead of 60 or 70%.

Answer to R2 Q2:

Thank you for the comment. We modified Figure 5b-d and Figure S10, in which the y-axis range was set from 0 to 150%.

Figure 5.

Figure S10.

3. Figure 6 shows a strong effect on xenograft growth of suppressing SHMT2 with a single shRNA, but concern for off targets in this xenograft experiment remain since only a single shRNA without a rescue experiment was used. This is of additional concern considering the relatively modest effects of targeting SHMT2 or treatment with SHIN2 in cell culture. Evidence of selectivity for ATC xenografts vs PTC xenografts might also strengthen the conclusions of this manuscript.

Answer to R2 Q3:

Thank you for your comment. We agree that additional experiments with PTC xenografts would help clarify our conclusion. Unfortunately, we could not perform such experiments within our timeline.

Minor points:

1. Please add the concentration of SHIN2 used for each experiment in all of the figures.

Answer to R2 Minor 1:

Thanks. We filled out the SHIN2 concentration in each experiment in all the figures.

Figure 5.

Figure 6.

Reviewer #3 (Remarks to the Author):

I thank and commend the authors for the thorough responses to the comments, the replies to which are satisfying to me. The quality of the single-cell data indeed seems very good.

However, regarding my first major comment on availability: I found that downloading the data available on GEO with the provided token only gave access to the gene counts, not to the raw data.

This is a sizable and important dataset for the field. The authors should make the raw data available, first for reproducibility per the Nature portfolio editorial guidelines; and second and as importantly, because its availability for reuse is critical to make this study of interest to a large audience.

Answer to R3:

Thank you for your comment. We have already uploaded all raw data (FASTQ files) for both bulk and single-cell RNA sequencing. However, due to GEO policy, the token does not grant access to the raw data. Rest assured, the data will be released to the public along with our publication.